# A Comprehensive Technological Survey on the Dependable Self-Management CPS: From Self-Adaptive Architecture to Self-Management Strategies

**DOI:** 10.3390/s19051033

**Published:** 2019-02-28

**Authors:** Peng Zhou, Decheng Zuo, Kun Mean Hou, Zhan Zhang, Jian Dong, Jianjin Li, Haiying Zhou

**Affiliations:** 1School of Computer Science and Technology, Harbin Institute of Technology, Harbin 150001, China; zuodc@hit.edu.cn (D.Z.); dan@hit.edu.cn (J.D.); 2LIMOS, UMR 6158 CNRS, University Clermont Auvergne, 63173 Aubière CEDEX, France; kunmean.hou@gmail.com (K.M.H.); jjli@isima.fr (J.L.); 3School of Electrical and Information Engineering, Hubei University of Automotive Technology, Shiyan 442002, China; zhouhy_dy@huat.edu.cn

**Keywords:** Cyber Physical Systems, Industry 4.0, MDE, lifetime verification & validation, dependability, correctness, flexibility, real-time self-adaptation, self-management, self-healing

## Abstract

Cyber Physical Systems (CPS) has been a popular research area in the last decade. The dependability of CPS is still a critical issue, and few surveys have been published in this domain. CPS is a dynamic complex system, which involves various multidisciplinary technologies. To avoid human errors and to simplify management, self-management CPS (SCPS) is a wise choice. To achieve dependable self-management, systematic solutions are necessary to verify the design and to guarantee the safety of self-adaptation decisions, as well as to maintain the health of SCPS. This survey first recalls the concepts of dependability, and proposes a generic environment-in-loop processing flow of self-management CPS, and then analyzes the error sources and challenges of self-management through the formal feedback flow. Focusing on reducing the complexity, we first survey the self-adaptive architecture approaches and applied dependability means, then we introduce a hybrid multi-role self-adaptive architecture, and discuss the supporting technologies for dependable self-management at the architecture level. Focus on dependable environment-centered adaptation, we investigate the verification and validation (V&V) methods for making safe self-adaptation decision and the solutions for processing decision dependably. For system-centered adaptation, the comprehensive self-healing methods are summarized. Finally, we analyze the missing pieces of the technology puzzle and the future directions. In this survey, the technical trends for dependable CPS design and maintenance are discussed, an all-in-one solution is proposed to integrate these technologies and build a dependable organic SCPS. To the best of our knowledge, this is the first comprehensive survey on dependable SCPS building and evaluation.

## 1. Introduction

The Cyber Physical System (CPS) concept was first proposed by the US National Science Foundation (NFS) in 2006. One year later, the President’s Council of Advisors on Science and Technology (PCAST) raised the CPS to ensure the continued leadership of the USA and recommended putting CPS as a top research agenda item. In 2012, German government put forward Industry 4.0 to develop Germany’s competitive position in manufacturing. Thereafter, the EU (Horizon 2020, in 2013), Japan (CPS Task Force, in 2015), and China (Made in China 2025, in 2015) published their own CPS/Industry 4.0 plans. Roughly speaking, CPS stems from the information and communications technology, and Industry 4.0 is based on manufacturing automation technology. They are two different evolutions with the same goal [1]. CPS integrates computation, networking, and physical dynamics [2,3], and has even been regarded as a next revolution of technology, which can rival the contribution of the Internet [4].

Besides CPS and Industry 4.0, lots of similar concepts have been proposed to describe the system from different perspectives [5,6,7]. Here, we just introduce the most similar technical concepts. From the viewpoint of architecture, there are Machine to Machine/Man (M2M) [8] and System of Systems (SoS) [9]. From the viewpoint of networked control, there are Wireless Sensor Actor/Actuator Networks (WSANs) [10] and Networked Control Systems (NCSs) [11]. From the viewpoint of communication and information processing, there are the Internet of Things/Webs (IoT/IoW) [12], fog computing [7,13], big data, self-adaptive control systems, etc. In this paper, we will use the term “CPS” to collectively denote such smart closed-loop systems.

With the rapidly increasing complexity, it is impossible for administrators to understand the massive complex data and then give proper commands in time to instruct CPS to take right activities. Thus, self-management (or autonomous computing) seems to be the only alternative solution. Some exploratory studies on improving some qualities of self-management, such as reconfigurability [14] and interoperability [15] and the QoS of services [16] have already been proposed. The concepts of self-* and autonomic computing have been proposed for a long time [17,18]. Many models and strategies have been proposed for building self-adaptive systems [19,20]. However, due to the limitations of current technologies, self-adaptation decisions are generally short-sighted, ineffective and unconstrained, which can’t well satisfy the safety-critical requirements. Thus, how to build highly dependable, controllable and predictable CPS is still an open issue.

CPS involves numerous multidisciplinary technologies. For example, smart manufacturing involves smart factory, IoT, smart sensor, big data, additive manufacturing, and holograms, etc. [6,21]. However, these technologies are usually studied separately, and few researchers discuss their interrelationships and integration in detail. Massive challenges still need to be overcame to build an organic, dependable SCPS with these fragmented technologies [22,23]. Among these challenges, guaranteeing the dependability of self-management is an urgent issue. Systematic solutions are necessary to develop a dependable self-management CPS (SCPS), which includes dependable and flexible architecture design, creatively integration of current technologies, and strictly dependability verification. Moreover, a dependable SCPS should be able to automatically evaluate the dependability and the risk of the self-management strategies at runtime.

According to the goals of adaptation, the self-adaptation of SCPS can be classified into two types: one is *environment-centered adaptation (self-adaptation)*, whose target is the external systems (e.g., physical world and humans); it is aimed at interacting with the changeable environment properly. Another is *system-centered adaptation (self-healing)*, which is aimed at guaranteeing the dependability of the cyber space (e.g., the reliability of infrastructures and the availability/quality of services). For environment-centered adaptation, SCPS should guarantee the fitness and safety of self-adaptation decisions, as well as the dependability (safety) of the adaptation procedures. For system-centered adaptation, SCPS should automatically diagnose the faults, remove or isolate the failures, and adjust its structure and behavior to keep the system healthy. In the real world SCPS, *these two kinds of adaptation interfere with each other, and even conflict with each other in some cases* (e.g., resource competition, and the tradeoff between energy budget and redundancy). It needs great wisdom to design a dependable SCPS, which can maintain the dependability by itself and smartly balance the dependability with other requirements in different scenarios.

In this paper, we use the term “self-adaptation” to represent the environment-centered adaptation, and the dependability of self-adaptation is mainly about the correctness and safety of decisions. And we use the term “self-healing” to represent the system-centered adaptation, which focuses on improving the dependability of the platform of CPS. “Self-management/self-management” is an integrated concept which includes “self-adaptation”, “self-healing” and more other concepts, such as self-protecting and self-optimization.

### 1.1. Motivation and Goal of This Survey

One can never emphasize too much the importance of the dependability to a safety-critical CPS. Because: (1) with the increasing complexity of systems, it becomes more and more difficult to evaluate the dependability during the design period. (2) CPS has to continuously and autonomously adapt to the changeable environment almost in real-time, which introduces serious challenges to runtime dependability V&V. (3) The effects on physical space can hardly be eliminated. CPS should carefully evaluate the risk of the decisions and guarantee the safety of the decision processing. (4) Whether subsystems fail or not, activities should be started at the right time in the right place, and processed at the proper speed. Hence, it urgently needs systematic solution to guarantee the dependability of CPS at both design period and runtime. Otherwise, CPS would not simplify our life but make the life tougher and more dangerous. A comprehensive technological survey is needed to guide the further research and improve the dependability of CPS.

There are 7948 papers are published on the topic of “CPS or Industry 4.0” in more than 115 fields according to a “Web of Science” search on 30 August 2018 (the detailed results are shown in Figure A1 and Figure A2 in Appendix A). However, publications on the dependability of CPS are few. Only 456 papers correspond to the topic of “dependability or reliability or availability” (detailed results are shown in Figure A3 and Table A1 in Appendix A). Upon further review, only 67 papers discuss the dependability in one or more sections. A similar conclusion can be reached for the survey on component-based CPS architecting since 2015 [24], where only nine of 1103 publications discuss reliability, and six papers are about maintainability. Currently, most surveys just mention the importance of dependability to CPS, and only one survey focused on the challenges of dependable infrastructures of CPS and discusses the dependability in detail [25]. With more and more researchers paying attention to dependability (as seen in Figure A3), a comprehensive survey of dependability solutions to guide the future research of CPS is urgently needed.

### 1.2. Literature Search Rule

In this survey, we focus on the dependability of CPS and search the papers with four groups of keywords on Web of Science, ACM Digital Library, IEEE XPlore, Springer Digital Library, Elsevier Science Direct and Wiley Online Library. The first group of keywords are the similar concepts of “CPS”, the priority of terms is illustrated as follows: “*Cyber physical System*” = “*Industry 4.0*” > “*Internet/Web of things*”(IoT/WoT) = “*Fog computing”* > “*System of System*” (SoS) > “*Machine to Machine*”(M2M) > “*Wireless sensor network*”(WSN) = “*Wireless sensor actuator network*” (WSAN) > “*networked control system*” > “*embedded system*”, which are denoted by *key_A*. The second group of keywords, *key_B*, are about the self-* characteristics of CPS, which include “*self-adaptation*”, “*self-adaptive*”, “*self-adapting*”, “*self-healing*”, “*self-maintaining*”, “*self-management*” and “*autonomic computing*”. The third group of keywords, *key_C*, are dependability-related, and include “*dependable/dependability*”, “*reliable/reliability*”, “*maintainability*”, “*fault-tolerance*”, “*safety*”, “*fault detection*”, “*fault diagnose/fault diagnosis*”, “*fault prediction*” and “*fault prevention*”. We searched the papers with the combination of “*key_A and key_B and key_C*”. Meanwhile, SCPS can also be regarded as a large scale real-time self-adaptive system. We also used “*real-time and key_B*” as the fourth group of keywords to investigate the self-management strategies. Moreover, we searched the papers on dependable self-management strategies with the combination of “*real-time and key_B and key_C”*. We eliminated duplicates and checked the abstracts to select the most relevant papers. Finally, we cited 240 papers and nine books among 1376 papers.

We note that SCPS involves massive interdisciplinary technologies. To build dependable SCPS, it needs systematic architecture design and elaborate integration of technologies. Generally this needs a long introduction and comprehensive analysis to clearly introduce the detailed solutions. Hence, we prefer to cite full papers rather than related short papers. To cover the advanced self-management strategies, more conference papers are referenced. In addition, we prefer to cite the studies on theory and practice rather than system introductions, and papers on systemic solutions rather than concepts, if they share the same topic/domain. ESI/SCI papers and highly cited papers are cited first.

### 1.3. Structure of This Survey

To build a dependable SCPS, it is necessary to co-design and co-evaluate the applied architecture and strategies. This work tries to provide a detailed, comprehensive investigation on dependability engineering and life cycle maintenance for SCPS. As SCPS is built with the existing (embedded) systems, to avoid unnecessary duplication of work, we assume that readers have a good background in the dependability, control system and embedded system. In this survey, we will focus on the new technological challenges of SCPS modeling, dependability verification & validation (V&V) and runtime dependability management. To simplify, we regard the subsystem as the unit and focus on the dependable integration among subsystems, and the dependable interaction between the cyber space and the physical space. This survey will discuss the dependability issues at three levels: (1) the self-adaptive architecture (including architecture and model based V&V), (2) dependable environment-centered adaptation (self-adaptation), and (3) system-centered adaptation (self-healing).

The rest of the paper is organized in an order of SCPS engineering (dependability requirement analysis, SCPS architecture design and Model based V&V, and system maintenance), as illustrated in Figure 1. It first recalls the concepts of dependable SCPS, and then proposes the generic processing flow, analyze the error sources and challenges of dependable SCPS engineering and runtime management in Section 2. We survey the state of the art in architecture design of SCPS, summarize the shortages of current architectures, and then propose a conceptual dependable self-adaptive architecture and the related technologies of improving the dependability of architecture in Section 3. We split self-management solutions into two sections, and discuss the self-adaptation technologies in Section 5 and self-healing technologies in Section 6. In Section 7, we complete the jigsaw of technologies and discuss the technical trends of dependable SCPS. An all-in-one conceptual solution is proposed for future SCPS development and maintenance. In Section 8, some interesting features of SCPS are discussed; we also conclude the solutions for 9 challenges which proposed in Section 2.

## 2. Background and Overview of Challenges of Dependable SCPS

The increasing complexity is a fundamental challenge to both SCPS design and maintenance. To design a dependable SCPS, we need efficient model theory and high fidelity model based engineering (MDE) toolset. And to simplify maintenance, we need a self-adaptive architecture, dependable self-adaptation strategies and self-healing solutions. In this section, we first recall the concept of dependability and the capability of dependable SCPS. Then we introduce the generic self-management flow of dependable SCPS. Next, we analyze the error sources of SCPS with a formal self-adaptation flow. Last, we introduce the role that the feedback mechanism plays in the self-healing solutions. Finally, we present the challenges of CPS engineering (modeling and V&V).

### 2.1. The Methodology for Dependable SCPS Engineering

Dependability is an integrating property, which encompasses three concepts: (1) the threats, which consist of errors, faults and failures; (2) the attributes, which include reliability, availability, safety, confidentiality, integrity, and maintainability; (3) the means to achieve the dependability, which contain fault prevention, fault tolerance, fault remove, fault forecasting. More detailed introduction on dependability and its threats and attributes refer to [26]. A dependable system should be able to notify the administrators of the risk of permanent faults, help administrators to identify the sources of errors and locate the failed subsystems. Moreover, the dependable system should be able to recover from the transient faults, isolate and tolerate the permanent faults.

However, CPS is so complex that no administrator can clearly understand its behavior and identify the symptoms of faults in time, let alone make proper management decisions. To simplify management, one promising solution is *applying self-management solutions (a.k.a. autonomic computing) to tame the complexity of management*. Self-management can be subdivided into *self-adaptation, self-healing, self-configuration, self-organizing, self-synchronization, self-protection*, *self-learning* and *self-optimization* [19,20,27,28]. Through self-* solutions, SCPS can automatically deal with the internal and external changes, and maintain the quality of services. However, complexity has negative impact on dependability. To reduce the complexity introduced by self-management services, we should systematically design the architecture and the interoperation interfaces, to facilitate self-management. Moreover, we should build a SCPS that can aware its statuses, model itself and the runtime requirements of self-management activities, so that SCPS can guarantee the quality of self-management and the dependability of activities.

Building a dependable SCPS needs to comprehensively integrate various technologies at different levels. *At the infrastructure level*, CPS needs more reliable, stable hardware and software to tolerate the interference from natural environment. *At the subsystem level*, CPS should be able to diagnose or predict the failures automatically, because there are too many subsystems for supervisor to detect one by one. *At the service level*, we need to develop high available solutions to provide 24/7 service supporting. *For decision making*, CPS should first provide integral and consistent information for Decision Support System (DSS), and then evaluate the fitness (i.e., correctness and safety) of decision at runtime. *For decision process*, CPS should select the most reliable services, and guarantee the reliability of commands and the safety of the process of decision. *For daily maintenance*, we need to improve the flexibility, observability, traceability to simplify the failure location and manual recovery. *Furthermore*, considering the large potential value of data, it is necessary to guarantee the confidentiality of information in the CPS. In this survey, we mainly focus on reliability and availability of architectures for SCPS, safety and maintainability of self-management at runtime.

As errors are accumulated with the procedure of self-management loop, undependable self-management will lead to more serious damages. It is necessary to verify each step of the self-management loop, which implies that we need to develop additional services to monitor and verify the self-management services. As we all know, complexity negatively affects the dependability. Therefore, it is necessary to reduce the side-effect of introducing self-management and to simplify the daemon services of self-management. To tame the complexity, one key rule is *using simplicity to control complexity* [29]. For SCPS, we should use relatively simple self-management services to control complex normal functions of CPS, and use simpler daemon to monitor self-management services. To achieve this goal, we need to find the universal schemas of CPS’s behavior and the related self-management services. In case of improper decisions, we also need to improve the observability, traceability and maintainability of SCPS. Thus, the administrators can supervise the CPS, fine-tune the strategies and take over the system in emergencies to avoid catastrophes.

### 2.2. The Process Flow of Dependable SCPS

CPS is a kind of closed-loop system with multi-actors, which include sensors, actuators, computing system (such as data collectors and analyzers, and decision support system), and communication networks, and even human beings [5,21], yet, few publications have discussed the influence of physical space/world on the cyber space in detail. Without the physical space, the closed-loop of interaction is incomplete. *To interact dependably, SCPS should not only consider the behavior of cyber space, but also take into account the physical effects (on the cyber space), even the effects of human beings*. Imagine that a man drives a car in terrible cold weather. The low temperature will not only slow down his reaction time, but also reduce the reliability of the car (i.e., reducing the friction between the tire and ground, weakening the strength of materials, etc.). A dependable self-driving system should be aware of the possible effects of cold weather and adjust the related weights to make proper decisions. For another example, after water crossing and wading, the braking effect becomes weak and the braking distance increases. A self-driving car should be aware of the change of braking distance and adjust the related parameters (such as the time to start braking). Furthermore, the dependable self-management should be perspicacious and far-sighted, so that the actuators can deal with the changes.

To adapt to the changeable environment, SCPS has to deal with uncertainties and should be able to tolerate unexpected failures at runtime. In detail, SCPS should be able to quickly sense the environment, be aware of its context, and predict the future, so that it can adjust its behavior to adapt to the environment or change the environment to protect itself (i.e., heating the battery in a cold environment). To cooperate with humans, SCPS should guess the intentions of humans and pre-process information for the next interaction(s). Meanwhile, to coordinate subsystems and to avoid over-operation, it should form a complete, sensitive and soundness closed reaction loop, so that the SCPS can adapt its behavior if the environment changes or the humans change mind. From this point of view, we propose a generic processing flow of environment and human in-the-loop SCPS, which is illustrated in Figure 2. The core participators in cyber space include sensors (network), actuators (network), networks and DSS. Sensors and actuators are the interfaces between cyber space and physical space. The physical system includes the devices of sensors and actuators, the human beings and the physical environment around them. We don’t take the DSS as a part of physical space because they are normally deployed in the ideal environment, which are barely affected by the physical/natural world. In this figure, the blue arrows represent the data flows of traditional CPS. The yellow arrows stand for the effects of the physical environment on the human and the devices. The green arrows show the flow of the status information about infrastructures (e.g., sensors, actuators and networks), which is very important to evaluate the dependability of SCPS. Dependable SCPS should monitor the status of physical systems (green arrows), and evaluate the effect of physical systems (yellow arrows), and guarantee the dependability of traditional process flow (blue arrows). To autonomously deal with the changes and failures, SCPS should form multi-level and multi-term feedback loop (which will be discussed in detail in Section 3.3), and overcome the limitations of subsystems, such as limited resources of embedded subsystem and lagging information of DSS. Meanwhile, the subsystems should also be smart enough to deal with the emergencies when the advices are not available.

Notice that the status of infrastructures also include the physical information, i.e., temperature of boards, the Relative Strength Index (RSI) of communication channels. DSS should take the physical effects on the infrastructures into account during decision making. And for human-in-the-loop CPS, DSS even should consider the intentions of human beings, reserve enough response time when interacting with human beings. As there is few efficient theory to model the human behaviors now, we will not discuss this topic in detail in this survey.

### 2.3. Formal Processing Flow of Self-Management and Error Sources

The formal self-adaptation flow is illustrated in Figure 3. $S_{c}(t)$ and $S_{p}(t)$ are the status of cyber space and physical space at time t, respectively. $S_{c}^{m}(t)$ and $S_{p}^{n}(t)$ are the minimum redundancy/maximum relevance observed status of the cyber space and physical space, where $S_{c}^{m}(t)\subseteq S_{c}(t)$ and $S_{p}^{m}(t)\subseteq S_{p}(t)$. The inputs for DSS are the sequences $\left(S_{c}^{m}(t-j),\cdots ,S_{c}^{m}(t)\right)$, $\left(S_{p}^{n}(t-i),\cdots ,S_{p}^{n}(t)\right)$ and the a priori knowledge. $D_{c}(t)$ and $D_{p}(t)$ are the self-healing decisions and self-adaptation decisions that are made by DSS. $\Delta S_{c}(t)$ are the corresponding self-healing activities of $D_{c}(t)$, and $⊕\Delta S_{c}(t)$ represents the effect of self-healing activities. Likewise, $\Delta S_{p}(t)$ represents the self-adaptation activities of $D_{p}(t)$ and $⊕\Delta S_{p}(t)$ is the effect. $⊕(\int_{t}^{t+1}f_{p}(t)dt+c)$ is the inertia effect of the physical phenomenon and ⊕ is a non-linear operator. The status $S_{c}(t+\Delta t)$ and $S_{p}(t+\Delta t)$ can be changed by the self-management activities, the inertia of physical processes and human beings, etc.

No matter whether we model the self-management loop as a linear control system $\overset{•}{S}=A(t)S+B(t)u+n_{e}$ or a nonlinear control system $\overset{•}{S}=f(S,t,u)+n_{e}$ (where $S=S_{p}^{n}\in {ℜ}^{n}$ or $S=S_{c}^{m}\in {ℜ}^{m}$, $u\in {ℜ}^{k}$, $n_{e}$ is the error), we still suffer from some hard issues caused by errors $n_{e}$. Generally, $n_{e}$ is modeled as a random variable, but in a real world system, $n_{e}$ depends on the error of observation values and the related operating functions, as well as their weights. Due to the limitations of current technologies, it is impossible to collect complete and accurate features, and it’s also impossible to analyze the abundant data in time. In practical engineering, we have to use $S_{p}^{n}(t)$ as the approximations of $S_{p}(t)$. Moreover, as the environment continuously changes, SCPS should dynamically adjust the weight matrices (*A, B*), the selected statuses $S_{c}^{m}$ and $S_{p}^{n}$, their dimensions (*n, m*), and the length of the historical status sequence (*i*), as well as the control rules u. Generally, we have to add additional a priori knowledge, such as high-level rules or domain knowledge base, to instruct the SCPS to recognize the different scenarios, and help the SCPS quickly choose the optimal parameters and control rules because no existing theory can recognize the changeable context from the mass of data, and automatically search the corresponding optimal parameters in time. Even if we had developed such a theory, it is still difficult for SCPS to accurately eliminate all the errors $n_{e}$. Moreover, SCPS has to search the sub-optimal combination of functions (or parameters) and generate defective decisions in a limited time Δt (notice that, Δt maybe a variable value). Such a compromise also inevitably increases the risk of failures.

In a nutshell, the error sources (issues) of self-adaptation are as follows: (1) *Long-term predictions are unreliable*, because the physical effects on the reliability of infrastructures are nonlinear (a.k.a. The Butterfly Effect). (2) *Status for decision making are incomplete*, $S_{c}^{m}(t)\subseteq S_{c}(t)$
*and*
$S_{p}^{n}(t)\subseteq S_{p}(t)$. (3) $S_{c}^{m}(t)$
*and*
$S_{p}^{n}(t)$
*are not trustable*, because the sensors may fail and data transmission may introduce errors. (4) *The timestamps of every sub-status (event) are not completely the same*. As the clocks of subsystems are not identical, we have to use the set $S_{p}^{n}(t)=\{s_{p}^{1}(t^{1}),\cdots ,s_{p}^{n}(t^{n})\}$, whose time is $t^{1}\approx \cdots \approx t^{n}\approx t$. This means that DSS may have different orders with original events, and the causal relationship of events may lose. (5) *Short-term decisions may be invalid or easy to miss the deadline*. Plans always fall behind the changes, so the predicted future is not identical with the real future $S_{p}(t+\Delta t)\neq S_{p}(t)⊕\Delta S_{p}(t)⊕(\int_{t}^{t+1}f_{p}(t)dt+c)$. What’s worse, and failures may have occurred during [t,t+Δt]. *The time maybe too short to find a self-healing solution, or even there may be no reasonable and practicable self-healing solution*. (6) *There is no effective theory to model the changeable factors (with the fixed matrixes*
$S_{p}^{n}(t)$
*and*
$S_{c}^{m}(t)$). Generally, the selected Key Performance Indicator (KPI) factors and their weights are not fixed. Fast context-aware solutions are necessary to search the optimal set of factors and weights. Further survey on dependable real-time self-adaptation will be discussed in Section 5.

### 2.4. The Schemas of Feedback Loop for Self-Healing and Self-Reference Problem

Self-healing is an important capability of the dependable SCPS. Compared with the self-adaptation, self-healing suffers from not only the same issues caused by errors, but also the self-reference problem. Generally, the feedback schemas of the self-healing solutions can be classified into two types, which are illustrated in Figure 4. For schema (a), the self-healing measures are highly integrated with normal function, the component will adjust its behavior to improve dependability during executing, i.e., some dependable control mechanisms [30]. For schema (b), the self-healing measures and the normal functions perform independently in space-time, i.e., expert system based self-healing. Compared to schema (a), schema (b) has higher flexibility and scalability, but it may react more slowly. Generally, schema (b) is more efficient in the large scale SCPS, and the causes will be explained in Section 3.3.3. Notice that, the manager in schema (b) is a component that contains the schema (a).

According to the occasion of taking healing measures, self-healing methods can be classified into two types: *preventive healing* and *remedial healing*. The preventive healing methods monitor the symptoms, predict faults, and then take actions to prevent the failures. These methods can be regarded as a special self-adaptation services which are dedicated to the health of system. In this context, *preventive healing methods suffer from all the issues that self-adaptation does* (as seen in Section 5.2), *but even more seriously* (because self-healing may also fail). Remedial healing tries to reprocess the services from some backup status after failures occur, and remove/isolate the error sources. As the recovery measures have to reprocess services, this increases the risk of missing deadlines. Moreover, due to fault propagation, it is very difficult to locate and remove the error sources in some cases. Consequently, similar failures may occur repeatedly and reprocessing may fail again and again. What’s worse, *self-healing suffers from the halting problem* [31] (which is a problem caused by self-reference, and it is an unsolvable logic puzzle). As seen in the schema (b), we also need to guarantee the dependability of the manager (because self-healing is also a service). For this purpose, we have to introduce a self-reference service (a.k.a. the service likes schema (a)), otherwise the self-healing chain is unclosed and some services can’t be healed.

### 2.5. The Challenges of Guarantee the Dependability of SCPS

Due to the increasing complexity, it is impossible to enumerate all scenarios and test all cases during the design period. What is worse, the traditional fault management strategies are dedicated to the particular organization in specified scenario, which are unable to effectively handle the failures caused by the dynamic behaviors. In this sub-section, we summarize three new challenges of (model based) dependability evaluation of self-adaptation and the relative solutions will be surveyed in Section 4. We also propose six new challenges of runtime dependability management of SCPS, the available self-healing solutions will be investigated in Section 5.

#### 2.5.1. The Legacy Issues

Though dependability has been researched for years, several challenges are still unresolved: (1) The harder problem causes by the increasing complexity, i.e., Testing is NP-hard [32], which implies that the published system inevitably contains bugs. (2) The cost of development increases exponentially with the degree of reliability [29], which implies that it is unpractical to improve the reliability of all subsystems. (3) Self-healing (self-detection) of Turing machine systems is a halting problem [31], which is a famous paradox in computability theory. In short, it is impossible to design a generalized algorithm (for Turing machines) which can determinately find out whether the program halts (fail) or not. The halting problem implies that it is impossible to achieve absolute dependability with self-healing solution. Hence, manual interventions are still necessary for SCPS. Nevertheless, self-healing can simplify the manual management and reduce the risk of misoperation. Otherwise, managing a complex system like SCPS will be a disaster.

For SCPS, dependability related issues are tightly intertwined with correctness issues. For example, unreliable data may mislead SCPS to make wrong self-healing decisions, and wrong decisions will reduce the dependability or even cause terrible failures. As the traditional fault management strategies can’t evaluate the correctness/fitness of decisions, these strategies are unable to effectively deal with faults caused by the misbehavior. Take the redundancy strategy as an example, all homogeneous sensors suffer from the same interference from the environment, the redundant sensors can’t tolerate such failures. In contrast, the redundant (distributed) homogeneous sensors may generate inconsistent observations, which will trigger different even conflicting activities.

Moreover, for traditional systems, their structures are predesigned and fixed, or rarely changed throughout the whole lifetime. Systems can only adjust their logical behavior, for example, selecting a new logical branch (i.e., the if-else pattern), switching the state of next operation (i.e., the feedback control x(t+1)=Ax(t)+Bu(t)), or renewing the threshold/reference value (i.e., time series model $x(t+1)=\sum_{i=0}^{n}w_{i}x(t-i)$). Systems barely change the organization or the topology of the (physical/logical) components. For example, the Triple Modular Redundancy (TMR) system can decrease the connection degree (between the majority gate and the redundant modules) from three to two and then to one. But all modules and links are predesigned, no new module or (physical/logical) link is appended in the degraded or recovered system, while for SCPS, new subsystems and links may be added and removed dynamically.

The dynamically changing structures of SCPS introduce another big challenge to system design and V&V. As we all know, it is impossible to verify all possible structures at the design period or at the plan making period. As a compromise, SCPS has to search the suboptimal topology and select the proper candidates at runtime, and then reorganize them as a temporary but cohesive team. Furthermore, subsystems may quit and join the team at any time because of failures, recoveries or movements. For each time of reorganization, SCPS should evaluate the risk and dependability of structure changes. In other words, every subsystem should be able to model the context and evaluate the risk and dependability at runtime. To achieve these objects, all subsystems of SCPS should apply model@run.time methods to support *self-modeling* and *self-evaluation.* Moreover, all model@run.time methods should be finished in a limited time.

#### 2.5.2. Technical Challenges of CPS Modeling and Dependability Analysis

As the future is uncertain, SCPS has to model at runtime to reduce the uncertainties and generate several backup plans. Meanwhile, SCPS should check the statuses and properties of (heterogeneous) candidates, then select the most suitable implementers to optimize the plans. To achieve these goals, SCPS should model both the physical space and the cyber space, and predict the future environment and assign activities to the proper candidates. As well, SCPS should evaluate the failure risk and guarantee that the selected candidates can well satisfy the requirements of decisions. It needs efficient theories and complex engineering solutions to enable SCPS to automatically model the dynamic structures and the stochastic behaviors. It is also full of challenges to achieve optimal compromise between dependability and other requirements under uncertainties.


*MQ1: Collecting the error sources and their failure distribution in different scenarios*


A tremendous amount of investigation is needed to identify the error sources and the relative failure distributions in different scenarios. What’s worse, we don’t have the accurate distribution of most failures at the design period. Moreover, the distributions and their scale parameters may change with time at runtime, i.e., due to aging of hardware, the failure rate λ of exponential distribution increases with time.


*MQ2: Fault propagation between subsystems in different modes (state space explosion and the validity of state combination)*


Each subsystem has several normal modes (i.e., high speed mode and energy saving mode) and error modes (i.e., degrade mode and failure mode). For different combinations, the paths and the effects of fault propagation are completely different. It is impossible to verify all possible combinations. Moreover, some modes are incompatible with each other, but we can’t forbid the combinations of them by rules, because some combinations are invalid in some scenarios but are valid in others, and we can’t distinguish these scenarios through rules or enumerations (at the design period).


*MQ3: Modeling the dynamic structures and self-adaptation behaviors (triggered by multiple stochastic physical signs) and evaluate the failures caused by multi-errors*


Most modeling methods are dedicated to static structures [33]. Few methods, especially few formal methods, support to model the dynamic structures and the stochastic interaction behaviors among subsystems. Hence, it is an arduous task to evaluate the dependability and prevent the failures. For complex large scale systems, several errors may be triggered together and even several failures may occur simultaneously. It is necessary to evaluate the co-effect of these failures.

#### 2.5.3. Technical Challenges of Runtime Dependability Management of SCPS

For SCPS, self-healing and self-adaptation are highly interrelated and mutually reinforcing. To simplify, the dependability of self-management can be divided into two parts: (1) the safety of decisions and the fitness of arranged candidates, (2) the reliability of infrastructures and the dependability of the procedure of decision process. Despite the traditional issues, the SCPS still faces six new challenges:


*RQ1: How to obtain consistent observation sequences on all subsystems (in other words, how can all distributed subsystems reach a consensus on the timing behavior of events)?*


Temporal issues become extremely critical [34]. Reasoning needs the temporal information of events, different temporal orders of two events lead completely different conclusions of the (causal) relationships, wrong timing orders will mislead the DSS to make wrong decisions.


*RQ2: How to make safe decisions with incomplete, inconsistent, inaccurate data and disordered events?*


In the real world system, data may be incomplete, messages and events may be delay and disorder. SCPS should still be able to make safe decisions and take right activities.


*RQ3: How to continuously quantify the dependability of each subsystem under different contexts, especially in the unusual situation with rare evidences?*


The physical space changes continuously, while the cyber system generally is discrete. It is difficult for cyber system to seamlessly switch between two states. I.e. the reliability of subsystem s under the context $c_{1}$ at time $t_{1}$ is $p_{r}(s|c_{1}(t_{1}))$, and $c_{2}$ at time $t_{2}$ is $p_{r}(s|c_{2}(t_{2}))$, how to model the probability function during $t_{1}\to t_{2}$ as we can’t build a continuous model to describe $c_{1}\to c_{2}$?


*RQ4: How to keep consistent quality of service when a subsystem joins or quits?*


In other words, SCPS should guarantee the correctness and QoS of decision processing with different heterogeneous (but replaceable) subsystems under the unpredictable environment.


*RQ5: How to tradeoff between dependability and other requirements dynamically?*


For SCPS, the distributed subsystems with limited resources should achieve real-time execution with minimum failure risk and energy consumption. Searching the optimal self-management solution is a typical dynamic optimization problem (DOP) [35].


*RQ6: How to remove cumulative errors and continuously maintains the dependability of services, especially the self-dependability manager?*


As well-known as the Butterfly Effect, errors will accumulate with feedback loops. SCPS should remove the errors to keep the health of system. Making decision with unreliable evidence to remove errors suffers from the same problem of halting problem.

The *RQ1* challenge is about the dependability of infrastructures (especially the global reference time) and the reliability of data (events). *RQ2* focuses on evaluation the safety risk of decisions. The challenges of *RQ3*, *RQ4* and *RQ5* are about the runtime dependability evaluation for self-adaptation. *RQ6* is about the correctness and reliability of self-healing. These issues are intimately entwined with each other. Therefore to build a dependable SCPS, it needs systemic design and skillful compromise.

## 3. Dependable Self-Adaptive Architecture Design

There is no doubt that architecture design plays an import role in SCPS engineering. In some senses, the upper bound capability of self-management depends on the architecture. When we chose one architecture, it always means that we have chosen a set of potential self-management strategies and given up others, i.e., distributed detection schemas are unsuitable for centralized architecture. Meanwhile, a good architecture can also reduce the complexity of SCPS, facilitate the arrangement of decision process and simplify the runtime maintenance, which can holistically improve the dependability of SCPS. In this section, we investigate the approaches on architectures/frameworks of dependable SCPS in detail. Based on top-down analysis, we summarize the strengths and weaknesses of current SCPS architecture and propose a concept of hybrid architecture. Thereafter, we conclude about available technologies and methodologies to build a dependable SCPS. Finally, we summarize the shortages of current self-adaptive architecture. Here, we use the term “architecture” as the short of “architecture or framework”.

### 3.1. State of the Art of CPS Architecture Design and the Key Technologies

SCPS is an organic system with loose physical structure but compact logic organization. In the static view, all subsystems of SCPS are loosely decoupled and each subsystem could be regarded as an autonomous system. In the dynamic view, SCPS should make rigorous decisions and the subsystems should cooperate tightly to adapt to the intricate situations. To design such a SCPS, the architects should synthetically consider the intricate indexes, such as complexity, correctness, dependability, performance, scalability and flexibility, and trade off among these requirements, which is a multi-objective optimization problem. Moreover, these indexes have nonlinear relationships with each other. Any fine-tuning may have deep effects on the abilities of SCPS.

As mentioned earlier, for SCPS, dependability issues are tightly intertwined with correctness issues. To build a dependable SCPS, we need to systematically consider the correctness and dependability (C&D). As CPS contains both discrete subsystems and continuous subsystems, to design the dependable architecture for CPS, we should pay special attention on the correctness of integration and the dependability of interaction between the discrete subsystems and the continuous subsystems. After analyzing 97 approaches, we selected 20 real (not conceptual) CPS architectures and summarized the key methods on C&D in those approaches in Table 1. These architectures can be classified into three types: Service Oriented Architecture (SOA) based architecture [36,37,38], Multi-Agent System (MAS) based architecture [13,14,39], and other aspect oriented architectures like 5C architecture (5C stands for Connection, Conversion, Cyber, Cognition and Configure) [40], etc. In these architectures, the continuous subsystems are encapsulated into some special discrete subsystems, or special agents are developed to transform the signs and events between the two types of subsystems [40,41].

Generally speaking, SOA is mainly designed for the resource rich CPS with central manager system. It highlights the flexibility, composability and reusability of services, which can facilitate the dynamic service reorganization. However, *SOA is not suitable for all (subsystems of) CPS*. According to design principle, the services in SOA are *stateless* [42] and *location transparency* [43]. While *both the decisions and the actions in SCPS are state dependent and location aware*. Moreover, the interactions with physical space are safety-critical, which need high observability and traceability for monitoring and controlling. Stateless services are unsuitable for composing these interactions. Furthermore, the SOA services have little autonomy and all of them are controlled by the central manager (the services choreographer), which implies that the system can’t response to the accidents quickly. *Notice that the delay rapidly increases with the amount of involved services*. In summary, *SOA is more suitable for the stateless, delay-tolerant cyber services*, such as the long-term decision making services, and data services for the slow subsystems, i.e., human beings.

MAS is a popular architecture designed for larger scale distributed, decentralized SCPS. Every subsystem (agent) in MAS is an autonomous system, it can make its own decisions based on local context. But these decisions generally are very simple due to the limited resources. For MAS, swarm intelligence is the key technology to achieve self-management. However, high autonomy reduces the performance and controllability in somehow. The stochastic behaviors of autonomous subsystems also make it difficult to formally evaluate the effects of decisions, which reduce predictability and trustability. Moreover, the decentralized consensus solutions for MAS are still suffers the challenge of slow convergence rate [43], while decentralized consensus is the prerequisite to make dependable self-management decisions. Compared to SOA, MAS based decentralized solution needs much fewer resource and has higher robustness and faster response speed (though the response activities may not be optimal). Generally speaking, *MAS is more suitable for the large scale geographically distributed system, especially whose subsystems have limited resources, such as WSANs*.

Moreover, considering the cost and risk of developing a new SCPS, some aspect-oriented architectures are proposed to integrate the legacy systems into CPS. Generally, the developers assemble the computer systems or agents with legacy (industrial control) systems and connect them with industrial Ethernet. These aspect oriented architectures are generally dedicated to the special application (domain). Actually, the aspect oriented architectures are more the long term evolution solutions than the special architectures. Some proxy agents are designed to translate the protocols of legacy systems, and a new central manager system is integrated to coordinate these legacy systems. As the legacy systems are not well-designed for CPS, the central system can’t collect detailed enough information for decision making. As a result, it is also impossible for CPS to achieve fine grained control. Hence, *these architectures are not well suited for large scale, complex SCPS*.

Strictly speaking, SOA and MAS are two kind of abstractions at different levels. The subsystems in MAS can be built with SOA if these subsystems have enough resources. I.e. the entire data center can be regarded as a powerful subsystem of MAS, and the data analysis services can be implemented with the SOA architecture. From this point of view, the hybrid architecture integrated with SOA and MAS is a promising solution for SCPS, especially if these subsystems provide the stateless services. For state dependent services, we still need to improve the services by patching the state information. Another graceful solution is redesigning the services and encapsulating them with some formal compositional abstraction, such as the actor. Actor is a generalized timed FSMs model, which supports cascade composition. For more detailed introduction about actor, the reader is advised to refer to the book [63].

Besides flexibility and autonomy, controllability is another important requirement to the SCPS. An ideal SCPS (with hybrid self-adaptive architecture) should be able to grant the controllable autonomy to every subsystem. The autonomy should perfectly fit the role and ability of subsystems. The SCPS can adjust and revoke the autonomy when the situation changes. Technically speaking, *SCPS should decouple the architecture/topology control logic from the normal service logic*, and every subsystem should follow the same control specification. This is also the core design philosophy of Software Defined Networks (SDNs) [50] or programmable networks [64], which tame the complexity through *decoupling* and *abstraction*. From this point of view, software-defined architectures (SDAs) or architecture-based self-adaptations (ABSAs) [13,52,53] are promising solutions for SCPS. Both of them significantly expand the potentialities of adaptation, which is useful for overcoming the challenges from *RQ1* to *RQ6*.

### 3.2. The Methodologies to Design a Dependable SCPS

SCPS integrates multi-level technologies, and these technologies have various alternative solution, while the complexity of different integrations are quite different. To simplify the evaluation of the C&D of design, we need to select the most suitable technologies and organize them with right structure. Many exploratory studies have been done on this domain. Focusing on C&D, we analyzed the technologies/methods applied in the selected papers, which are cited in Table 1. These technologies/methods improve the C&D from diverse aspects.

#### 3.2.1. Reducing the Complexity (Benefit for *MQ3*, *RQ1*, *RQ3*, and *RQ4*)

*Decoupling* and *abstraction* are the two widely applied basic methods. Many successful cases have proved that their efficiency on taming the complexity and improving the flexibility and the maintainability. *Standardization* is another method to reduce the complexity of self-organization. It simplifies the orchestration of (heterogeneous) subsystems with concordant standards of behavior, such as the standard of interfaces and communication protocols. Decoupling and standardization are two native design philosophies, but can produce comprehensive benefits for architecture design, technology organization, and the strategies design. Though decoupling and standardization can’t solve the problems directly, but they can systematically relieve the modeling problem *MQ3* and the dependability management problems *RQ1, RQ3* and *RQ4*.

#### 3.2.2. Isolation and Migration at Task Set Level (for *MQ2* and *RQ4*)

*Virtualization* and *task migration* are two closely associated technologies, which are wildly applied in cloud-based CPS [45,47,49]. Thanks to the development of hardware, virtualization becomes an alternative solution for embedded systems [62]. There are two types of virtualization, one is virtual machine-based virtualization, and another is container-based virtualization. They respectively provide a virtualized, isolated runtime environment at the subsystem level and the application level. With virtualization solutions, the related tasks be organized as an isolated group and be migrated together between physical hosts. Through task migration, we can apply scheduling solutions at the task set level to balance the resource consumption among physical hosts and reduce the data access delay. It enables the developers to design scheduling strategies without considering the detailed constraints and dependency relationship among tasks. However, the prerequisite of task migration is that the tasks are location transparency, which limits the application scope to the data center (a.k.a. DSS). While to sensors and actuators, task migrated may be absurd. I.e. for the irrigation system, it is ridiculous to migrate the watering task from a dry place to a humid place. Compared with virtualization, *multi-role agent* is a finer-grained decoupling solution. Every subsystem of MAS can play one or several roles and even switch between different roles at runtime to meet the dynamical requirements of self-management, i.e., the actuator can play as a sensor to share its observation, or any subsystem can be a coordinator to coordinate the activities among neighbors.

#### 3.2.3. Enhancing the Dependability at Architecture Level (for *RQ5* and *RQ6*)

According to the analysis in Table 2, *temporal-spatial redundancy* is still the basic methodology to improve the reliability. Temporal redundancy technologies mainly include checkpoint and redo (log), which are mainly applied on DSS because of the time cost. To achieve spatial redundancy, virtualization is a new cost–effective solution to isolate the faults and prevent the error propagation. Task migration can improve the availability with warm standby or hot standby technologies. By using load balance technology, it can also reduce the risk of deadline missing and failure rates. Other spatial redundancy technologies include fault-tolerant protocols [65] and middleware [36], and multi-version, multi-copy (MVMC) [66], etc. Apart from the redundancy methods, we can also improve the dependability of SCPS with *design diversity* [66,67,68,69]. As an analogy with the importance of species diversity to the ecosystem, design is meaningful to SCPS. As it is impossible to test all code, the released system inevitably contains defects. It needs multi-version subsystems to tolerate the unexpected failures and improve the survivability of SCPS. Meanwhile, multi-objects decision making is NP-hard problem, diversiform optimization solution can increase the possibility to find the proper decision. However, some studies also show that diversity will increase the rate of undetected failures and design diversity takes positive effect only after certain quality threshold reached [68]. Considering the defect of redundancy based homogeneous solution (as seen in Section 2.5.1), we believe that *design diversity will an effective supplementary measure for building dependable SCPS*. One promising application of design diversity is to integrate different technical standards to compensate for each other’s shortcomings. For example, lidar, radar and digital cameras are integrated together to enhance vision of self-driving car in different weather. Different wireless standards are integrated to balance the power consumption and transmission distance, such as IoT node integrates with IEEE 802.11 (Wi-Fi), IEEE 802.15.1 (Bluetooth), IEEE 802.15.4 (ZigBee, 6LoWPAN), Sigfox, LoRa, WMBUS and NB-IoT [50,70]. Heterogeneous CPU are integrated to balance the performance and the energy budget, i.e., subsystem may be integrated with FPGA, GPU, general CPU and low power MCU.

#### 3.2.4. Improving the Quality with Formal Model and Formal Analysis Methods (for *MQ1* to *MQ5*, and *RQ2* to *RQ4*)

To be self-manageable, a SCPS should firstly model itself and analyze the status of both the cyber space and the physical space at runtime [71,72,73]. As shown in Table 2, we can design formal compositional architectures and formal quantifiable behavior models to improve the quality of SCPS, and to simplify the evaluation, especially the runtime self-evaluation, which can reduce the complexity of the model@run.time solution. It is a classic case of “using simplicity to control complexity”. Meanwhile, formal/mathematical model@run.time solutions can simplify the implementation, which can improve the correctness of analysis methods and reduce the side-effect of introducing model@run.time. Meanwhile, formal design overcome the drawbacks of design diversity, i.e., checking the consistency of behavior with isomorphic theory. And also we can apply compensation methods to keep the consistency of the nonfunctional requirements between different versions, i.e., using the reserved waiting time to align the execution time of two finite state machines.

### 3.3. Improve the Dependability of Self-Adaptive Architecture

Design a dependable self-adaptive architecture is the prerequisite to *overcoming the challenges RQ1 to RQ6*. A good self-adaptive architecture can properly integrate different technologies without introducing too much additional complexity, significantly widen the design space of self-management strategies, simplify the orchestration of services, facilitate the interactions between subsystems, clarify the causality of events, reduce the complexity of system evaluation, and boost the performance of self-management.

#### 3.3.1. Simplify Self-Management with Hybrid Self-Adaptive Architecture

The popular solutions of self-adaptation architecture include architecture-based approaches, multi-agent-based approaches and self-organizing-based approaches [74]. Due to limitations of current technologies, single self-adaptation architecture can’t meet well the requirements of large scale (geographically) distribution, smart and quick reaction. Obviously, most of subsystems, i.e., sensors and actuators, are not powerful enough to process data-driven decision making (i.e., machine learning-based technologies). It is also impossible to deploy all services on one single subsystem. Hence, we need to subdivide the role for subsystems and divide their tasks, which can improve the performance and reduce the global complexity (the detailed analysis is seen in Section 3.3.4). Meanwhile, it takes too much time to transmit data from sensors and actuators to the data center, which reduces the timeliness of decision making. The lagging decisions may not be able to catch up with the changes of the real world. Hence, neither decentralized architecture (i.e., multi-agent-based approaches) nor centralized architecture (i.e., SOA-based approaches) can meet well the requirements of intelligence and response speed. Hence, hybrid architecture with multi-term self-management is the only reasonable compromise, which is shown in Figure 5. To achieve both high intelligence and real-time reactions, we also need an efficient multi-level and multi-term feedback loop to achieve real-time self-adaptation, which is shown in Figure 6 (to simplify, only two levels of loops are shown in this figure). The global DSS can take full advantages of massive data to make long-term prophetic advices, and the local DSS can refine these advices and make best suitable decision to adapt to the local situation.

With hybrid multi-level architecture, the global DSS can use the big data analysis technologies and “AI” technologies to make wiser, more farsighted self-adaptation decisions. The powerful cloud system also make it possible to systematically analyze the safety of prophetic adaptation decision with multiple complex evaluation methods from different angles (as seen in the Section 5.2), then generate dependable advices to process the proactive decision. The local DSS can refine the constraints of advices with newest observations, select the proper candidates of decision executors and organize them in right order to process in the optimal way. For some emergencies, the actuators can take actions based the rules without waiting the decision from the DSS. With multi-level and multi-term feedback loop design, the SCPS can achieve high intelligence as well as real-time response. Meanwhile, the hybrid multi-level architecture can also tolerate the failure of partial subsystems, which improves the dependability of SCPS [75]. Through elaborated integration of decentralized and centralized architectures at different levels, the hybrid multi-level architecture can overcome the shortages of them and achieve high performance, scalability and flexibility. Meanwhile, hybrid architecture is also a long term evolution solution, which can well integrate the legacy systems.

#### 3.3.2. Guarantee the Timing Dependability of Events for Reasoning (for *RQ1*)

Guaranteeing the consistency of timing behavior is a systemic challenge for (large scale) distributed parallel systems [34,76]. To SCPS engineering, the timing dependability is more serious and urgent [34,40,77], because the applied reasoning methods are based on the analysis results of the causal relationships of events. Timing order is one key index of the causal relationships. If SCPS can’t reproduce the timing relationships of events correctly, it will generate wrong view of the statues of cyber space and physical space, and get wrong analytical results, which will mislead the DSS and self-healing manager, even lead to disasters in some cases.

The precision of timing depends on massive of factors, such as the quality of hardware (clock), the dependability of software, and the software control strategies. Due to the limitation of current technologies, i.e., the deviation between crystal oscillators, the stochastic jitter of signals, etc. [34,76], it is very difficult to achieve the precise global reference time (absolute time) in distributed CPS. What’s worse, timing has never been treated as a basic premise before [77], which implies that the legacy hardware and software systems can’t support well the requirements of precise timing. Without dependable information of timing behavior, it is impossible to generate dependable self-management decisions, let alone dependable SCPS. Fortunately, more and more researchers aware the importance of timing and try to redesign the whole system from hardware platforms [78] to operating system [79], from programming models [80] to architectures [81], and from time synchronization protocols [82] to communication standards [83,84,85]. These studies mainly focus on improving the precision of the timestamp of events, the timeliness of actions (a.k.a. taking actions in right time) and the stability of the time of communications and decision processes (i.e., decrease the difference between worst case execution time and best case execution time).

Timing dependability is an open issue to SCPS. With the increasing scale and autonomy, it becomes more and more difficult to reach a consensus about timing behavior. One natural idea is to build a new programming model with embedded temporal semantics [73,86]. The conceptual programming model is illustrated in Figure 7. For each task or decision, SCPS will generate a decomposable contract with timing and dependability requirements. Every subsystem should strictly follow the contract and process the related activities on time and in time. To tolerate unexpected delay, it can reserve some time for each activity. Meanwhile, several redundant subsystems are arranged together to process the decisions together to minimize the risk and improve timing dependability.

Take the case in Figure 7 as an example, there is a decision with the contract whose maximal process time is 50, and the reliability is greater than 0.97 (T < 50, and R > 0.97). For the subsystem $S_{2}$, its worst case execution time (WCET) is 3 and its reliability is 0.98, the decomposed contract for $S_{2}$ is (T < 6 and R > 0.97), the reserved time is 6 – 3 = 3. For the subsystem set $S_{1}$, as no single subsystem can satisfy the contract (0.8 < 0.97 and 0.6 < 0.97), m redundant subsystems are arranged to process the decision together. The results of redundant operations and parallel operations should be synchronized when they are finished (the synchronization is shown with dotted arrows, such as the input arrows of $S_{k}$ in Figure 7).

Though the contract based programming model can achieve high flexibility as well as controllability and predictability, it is not compatible with legacy systems. To apply this model, we should redesign all subsystems, also get the precise values of the best case execution time (BCET) and WCET, and narrow the range of [BCET, WCET]. Meanwhile, the model also needs precise real execution time. To satisfy this constraint, we should design precision timed infrastructures, which is still an open issue [76]. To get precise BCET and WCET, we also need platform dedicated tools (which will be introduced in Section 4), and the related methods to alleviate the time analysis problems that discussed in Refs. [87,88]. Moreover, we need effective runtime reliability evaluation methods to avoid misleading estimates, and to avoid applying wrong fault-tolerance scheduling strategies [89].

#### 3.3.3. Improving Composability and Compositionality of Services (for *RQ1* to *RQ6* and *MQ1* to *MQ5*)

To adapt well to the changeable environment, the SCPS should select the most proper subsystems, organize them with the right topology, and arrange them to process the decision in right order. To cooperate smoothly, subsystems should not only share the interoperable interfaces and identical data specification but also provide service in a consistent way, i.e., if subsystem A is replaced by subsystem B for some reason, B should not only continue the services, but also provide them with the same quality as A dose, or at least no worse than A. Only in this way can SCPS provide seamless services reorganization and process the decisions dependably. In other words, SCPS should evaluate the differences of properties between subsystem A and B, and guarantee that the replacement will not introduce any intolerable inconsistency (i.e., executing different if-else branches, because the conditions of decision may be near the critical value). To tame the complexity of the arrangement and related evaluation, we should improve the *composability* and *compositionality* (C&C) of subsystems and decisions [5,90,91], so that a complex decision can be decomposed into several small subtasks and processed independently; as well, as well, the global behavior can be keep consistent with the original solution. Meanwhile, the subsystems with good C&C can cooperate and switch smoothly, which is meaningful to maintenance.

*Composability* is the quality that component properties do not change by virtue of interactions with other components [91]. It originates from the philosophy of reductionism, which highlights the consistent behavior of the subsystem when it cooperates with other subsystems to build a larger system. On the contrary, *compositionality* is originated from holism. It is the ability that system level properties can be computed from and decomposed into component properties [91]. Compositionality is more about the capacity to decompose the system level properties. It focuses on the consistency between the system level properties and the divided properties (component properties), where the system level properties can be reasoned with the properties of components/subsystems. More detailed discussion about C&C refers to [91]. By the way, the concepts of C&C are interchangeable in some studies.

Improving C&C can symmetrically and significantly promote the quality of service composition, which has attracted more and more attention. As heterogeneity of abstraction layers leads to loss of predictability of system behavior, through adding additional semantic information of interactions, Sztipanovits et al. presented a passivity-based approach to decouple system stability from cyber timing uncertainties [92]. Focusing on the vertical composition, the approach explored the systematic development method for CPS integration [92]. Nuzzo et al. adopted contracts-based component specification and abstraction, and provided a platform-based design methodology with formal supporting for the entire CPS design flow [93]. Attie et al. proposed a general formal framework for architecture composability based on an associative, commutative and idempotent architecture composition operator; the authors established preservation of safety and liveness properties by adopting architecture composition [94]. Aiming at constructing a compositional proof of correctness, Sanjit A.S check name—does not look like a surname presented a formal methodology and a theoretical verification and synthesis framework integrating inductive learning with deductive reasoning [95]. Stavros detailed the key principles of compositionality focusing on interface design for MDD [96]. A compositional specification theory for components reasoning is proposed in Ref. [97], this specification theory uses synchronistic input and output (I/O) actions to abstract the temporal ordering of behavior. To ensure the interoperability of components and adherence to specifications, a generic algebraic method is developed and two strategies are proposed to synthesize or refine a set of contracts to evaluate the composition satisfaction of a given contract [98]. Moreover, the authors also presented a set of decomposing conditions for verifying the decomposition of a contract into a set of contracts [98].

Currently, C&C are mainly researched as a part of model driven engineering/development (MDE, MDD). Considering the similarities of the model at design period and at runtime, we proposed a formal decentralized compositional framework and developed a compositional actor based prototype system to explore the value of C&C to SCPS [73]. As SCPS is a kind of SoS, we believe that improving C&C of subsystems can comprehensively reduce the evaluation complexity of runtime composition and guarantee the C&D of self-management. Currently, two types of solutions are proposed to guarantee quality of composited service at runtime: one lightweight solution is to focus on the *invariant requirements* [57], another universal solution (as shown in Figure 7) is to build *formal mapping* and *calculation rules* between the requirements/contracts and the properties of subsystems [73]. A SCPS with good C&C can comprehensively, systematically simplify the (model@run.time based) evaluation of runtime composition and arrangement of candidates, and improve the quality of self-management. Overall, improving the C&C of SCPS is a key solution to overcome the challenges *RQ1* to *RQ6* and *MQ1* to *MQ5*.

#### 3.3.4. Improve the Dependability with SDA (for *MQ3* and Reducing Complexity)

As we all know, complexity negatively affects the dependability of system [29]. However, with the increasing scale of SCPS, we have to deal with the rapidly increasing complexity. As mentioned above, software defined solution, which is named as SDN or SDA or ABSA, is one promising approach to tame the complexity. In this subsection, we formally explain how to reduce the complexity and improve the dependability of system with SDA (a.k.a. decoupling the (architecture) control logic from normal functional logic).

In SCPS, we have three types of available organization, which are shown in Figure 8. Let us suppose that there are n subsystems $\left\{A_{1},\cdots ,A_{n}\right\}$. In the schema (c), there is one additional subsystem $A_{m}$ to manage the structure/topology control logic. To simplify, we assume that the reliability of all subsystems are the same, and just take the reliability of software into account. We adopt the commonly used exponential reliability function $R(t)=e^{-\lambda t}$, where λ is the failure rate. To simplify, we assume λ is proportional to the software complexity C, λ=kC where k is a scaling constant (*k* depends on the development effort) [29]; that is $R(t)=e^{-kCt}$. Let us normalize the mission duration t to 1 and let the scaling constant k=1. As a result, we can rewrite the reliability function with a normalized mission duration in the form $R(c)=e^{-C}$.

Let us denote $C_{i}^{f}$ as the complexity of function logic (business logic) of $A_{i}$ and $C_{i}^{c}$ as the complexity of control logic of self-organization. The complexity of $A_{i}$ in schema (a) is $C_{i}^{f+c}$ (business logic and control logic are not decoupled). Let us assume that the decoupling effects (decoupling and abstraction) are correct and effective.

Hence, we can reduce the complexity of system. Formally, we have $C_{i}^{f+c}>C_{i}^{f}+C_{i}^{c}$, (actually, software has at least polynomial complexity, i.e., according to the Agresti-Card-Glass system complexity metric [99], the global complexity of a software is $G_{t}=S_{t}+D_{t}$, where $S_{t}\;=\;O(\sum \left({\left(f_{i}\right)}^{2}\right))$ is the simplified Henry-Kafura information flow complexity metrics for modular without fan-in, while the original metrics is $S_{i}=length_{i}*{\left(f_{i}(in)*f_{i}(out)\right)}^{2}$ [100], $f_{i}(in)$ and $f_{i}(out)$ are the fan-in and fan-out of the module $A_{i}$; $D_{t}\;=\;(\sum V_{i}/(f_{i}+1))/n)$ is the mean of all the internal complexity measures for all of the modules, which is similar to the Halstead’s metrics [101], $V_{i}$ is a function of the amount of internal variables). In the schema (c), the control logics are moved to the manager $A_{m}$, $A_{i}$ keeps fewer logic to response to the commands from $A_{m}$. Notice that $A_{i}$ can reuse the communication code. Let’s assume that the additional complexity of response logic of $A_{i}$ is $C_{i}^{r}$. For a dynamic system like SCPS, we have $C_{i}^{r}<C_{i}^{c}$. The complexity of $A_{m}$ is $C_{i}^{c}$. Let us assume that $C_{m}^{c}\;=\;\sum_{i=1}^{n}\left(\beta_{i}*C_{i}^{c}\right)$, where $\beta_{i}$ is a scale factor that depends on the reuse rate of control logic of $A_{i}$, $\beta_{i}\in (0,1)$.

The reliability of $A_{i}$ in the schema (a) is $R(A_{i})=e^{{-C}_{i}^{f+c}}$, and the global reliability is $R(a)=\prod_{i=1}^{n}e^{{-C}_{i}^{f+c}}=e^{-\sum_{i=1}^{n}C_{i}^{f+c}}$. For $A_{i}$ in the schema (b), its reliability is $R(A_{i})=e^{{-C}_{i}^{f}}*e^{{-C}_{i}^{c}}\;=\;e^{{-(C}_{i}^{f}{+\;C}_{i}^{c})}$, and the global reliability is $R(b)=\prod_{i=1}^{n}\left(e^{{-C}_{i}^{f}}*e^{{-C}_{i}^{c}}\right)=e^{-\sum_{i=1}^{n}{(C}_{i}^{f}{+C}_{i}^{c})}$. The reliability of schema (c) is $R(c)=e^{{-C}_{m}^{c}}*\prod_{i=1}^{n}e^{{-C}_{i}^{f}{-C}_{i}^{r}}\;=\;e^{-(C_{m}^{c}+\sum_{i=1}^{n}{(C}_{i}^{f}{+\;C}_{i}^{r}))}$.

To improve the reliability: (1) As $C_{i}^{f+c}{\;>\;C}_{i}^{f}+C_{i}^{c}$, obviously, we have R(b)>R(a). (2) We have R(c)>R(b) if we can achieve $\beta_{i}\leq 1\;-\;C_{i}^{r}/C_{i}^{c}$ through decoupling. Proof. the inequality can be transformed into $\beta_{i}*C_{i}^{c}\;+\;C_{i}^{r}\leq C_{i}^{c}$, then we have $\sum_{i=1}^{n}\left(\beta_{i}*C_{i}^{c}+C_{i}^{r}\right)+\sum_{i=1}^{n}C_{i}^{f}\leq \sum_{i=1}^{n}C_{i}^{c}+\sum_{i=1}^{n}C_{i}^{f}$. With $C_{m}^{c}\;=\;\sum_{i=1}^{n}\left(\beta_{i}*C_{i}^{c}\right)$, we have $C_{m}^{c}+\sum_{i=1}^{n}\left(C_{i}^{r}+C_{i}^{f}\right)\leq \sum_{i=1}^{n}\left(C_{i}^{c}+C_{i}^{f}\right)$. Because $R=e^{-C}$ is a strictly monotone decreasing function, we have R(c)>R(b). As self-healing is also a type of control logic, external self-healing is recommended for complex SCPS (conclusion for Section 2.4). Notice that, $\beta_{i}\leq 1\;-\;C_{i}^{r}/C_{i}^{c}$ is a strong condition. In practice, it is acceptable that some subsystems don’t meet the condition.

From the view of structure/topology reorganization, most subsystems of SCPS, especially these redundant subsystems, have similar control logics/operations, i.e., finding the best successor, disconnecting, waiting and synchronizing. With the increasing of the scale, SCPS has more similar subsystems and it is easier to have small $\beta_{i}$. The reliability equations show that it can improve the reliability of SCPS by decoupling the similar control logics and moving them to a special manager subsystem. SDA can simplify the management of version consistency and improve the utilization of resource. Moreover, SDA can also reduce the differences in behavior that caused by diversity, which can improve the stability and predictability. The greatest advantage of SAD is that we can design a formal programmable contract, and the manager $A_{m}$ can adjust the behavior of subsystems according to the contract. Furthermore, the coordinator subsystem can generate a new service to coordinate the temporary team. As most of contract validation are the same, i.e., the grammar checking, and the compilation (if we use bytecode supported language like the java), the integration of these codes can significantly improve the dependability of contract and simplify the normal functional logic.

### 3.4. Summary of the Dependable SCPS Architecture and Organization

In this section, we survey the technologies/methodologies to design a dependable self-adaptive architecture. Obviously, it is impossible to achieve the dynamic adaptation in a changeable environment with static architecture and predefined rules. Hence, we need to design a dynamic and flexible architecture for SCPS. With a dynamic architecture, SCPS can generate the optimal solution for each adaptation decision by selecting the best subsystems and arranging them with an optimal organization. For engineering, we need to follow the methodologies of decoupling, abstraction and design diversity, design a formal compositional architecture, and improve the C&C of heterogeneous subsystems. Therefore, we can be able to build a dependable SCPS.

Reducing complexity and using simplicity to control complexity are the core methodology to improve the dependable SCPS at architecture level. *Decoupling* and *abstraction* are the cornerstones of these technologies and are wildly applied in architecture design. With the increasing scale of SCPS, organization management becomes more and more important. *Decoupling the control logic* of self-organization from functional/business logic is an inevitable trend, such as ABSA and SDA. Considering the limited resources of a single subsystem, the roles of subsystem will be subdivided and subsystems just focus on partial tasks, such as fault detection and fault diagnose. The architecture with *multi-role* subsystems will be a natural solution to SCPS. Hence, we can design heterogeneous subsystems with specified infrastructures (hardware and software) to perfectly match the requirements of roles.

Redundancy and fault isolation are the two basic solutions to improve the dependability of system. For the subsystems with rich resources (i.e., DSS), virtualization is a cheap and efficient solution to implement them. For the sensors and actuators, the common solution to achieve fault tolerance is deploying redundant physical devices. Another popular method is design diversity, which is adopted to tolerate the failures that caused by the unexpected environment and defect of software. Task migration/scheduling is the basic technology for self-healing (i.e., Fault recovery and fault avoidance). Currently, rules, knowledge and ontology are three mainstream methods to implement self-management decision making. Data driven methods, especially machine learning based methods, are becoming more and more popular for self-adaptation. However, the researches of data driven self-healing are few. One main reason is that current data analysis theory is not appropriate for analyzing the causality, and more detailed discussion will be found in Section 6.

Redundancy with the same subsystems can’t avoid the failures caused by the defects of design. To tolerate the unexpected failures and improve the survivability of SCPS, we need to apply diversiform design. We believe that *design diversity is the inevitable trend* for SCPS engineering. However, design diversity increases the inconsistency of the heterogeneous subsystems. It is recommended to *apply design diversity with standardization and formal abstraction*. Standardization can provide consistent definition of interfaces and message format, and formal abstraction can generate diversiform but isomorphic subsystems, which are two useful solution to avoid ambiguous behaviors. *Isolation* is another remedial measure to reduce the effect of ambiguous behaviors.

To improve quality of services, the nonfunctional requirements should be quantifiable and decomposable, so that subsystems can clarify their roles and responsibilities. The alternative solutions are *improving the C&C* of nonfunctional requirements and *building quantitative evaluation functions* between requirements and the properties of subsystems. Only in this way, will SCPS be able to decompose the decisions and arrange the best suitable subsystems to process them; and can the subsystems perform their own duty and meet the global requirements. However, the C&C of legacy systems generally are not very good. To build a dependable SCPS with legacy systems, we need to redesign or repackage the legacy systems to improve their C&C.

We believe that good C&C is a necessary feature for SCPS and all subsystems. Because a subsystem with good C&C can smoothly interact with other subsystems. A decision with good C&C can be easily decomposed to sub-decisions and processed accurately by subsystems. We also believe that formal architecture is a good solution to improve the C&C of SCPS. It can simplify dependability V&V of design and the risk evaluation of self-management decisions. Moreover, formal architecture is also an efficient solution to achieve high C&C. It can simplify the V&V of design and the evaluation of decision, and fault detection and diagnosis. What’s more, formalization can also improve the controllability and predictability of the behavior of SCSP.

## 4. Guarantee the Dependability of the Design with Model Based V&V

There is no doubt that models play important roles in both SCPS design and decision evaluation. With model-based dependability V&V, we can point out the vulnerable subsystems and remove the defects as earlier as possible. Dependability models have been widely researched at different levels. However, these models generally focus on dependability of partial function and can’t well evaluate the failure risk of the interaction between the technologies in different disciplines, as well as the co-effect of cross-layer technologies. Moreover, dependability solutions also introduce additional complexity, it is necessary to evaluate the efficiency and side effects of these solutions. Meanwhile, the models and V&V methods can also be applied for model@run.time evaluation to guarantee the C&D of self-management decisions. In this section, we review the current researches on model based dependability V&V of SCPS (dynamic system). Thereafter, we analyze the challenges of the current modeling tools and propose a concept solution of MDE based dependability V&V.

### 4.1. Current Researches on Model Based Dependability V&V for SCPS

SCPS is safety-critical, we cannot emphasize the importance of dependability of SCPS too much. There are massive of dependability V&V models/methods, such as Fault Tree Analysis (FTA), Binary Decision Diagram (BDD), Reliability Graphs, Failure Modes and Effects Analysis (FMEA), Failure Modes Effects and Critical Analysis (FMECA), Hazard and Operability Analysis (HAZOP), Functional Hazard Assessment (FHA), Risk Reduction Analysis (RRA), Markov chain analysis, Petri Net analysis etc. However, the traditional methods are designed for the system with static architecture, which are not powerful enough for evaluating the dependability of SCPS. In this survey, we focus on the dependability evaluation methods for dynamic system (SCPS can be considered as a kind of dynamic system), and the detailed introduction of traditional dependability V&V methods refers to [102,103,104,105,106,107,108,109].

To analyze the random failures of the dynamic behavior, simulation model seems the only effective V&V method. Generally speaking, there are three types of solution to build a simulation model. One is building the model with general purpose programming language, i.e., the network simulators (NS3), which is the most powerful solution. However, *with the increasing complexity, it becomes more and more difficult to evaluate the correctness of the model itself*. The second solution is extending the traditional dependability models by appending dynamic temporal semantics and structure description semantics. For example, a formal modeling approach with object-oriented Petri nets is proposed for High-Confidence CPS [110]. However, *the extended models are still not powerful enough to analyze the detailed adaptation behaviors*. These solutions can’t model the behavior that one subsystem joints the leaves the makeshift team, and it is also difficult to model the correlation distribution between inputs and outputs.

The third solution becomes more and more popular, it tries to build dedicated dependability models with more powerful formal languages, and the languages should support to model the dynamic behavior, i.e., high level dynamic formal language [111] (such as DEPICT [112] and Architecture Analysis & Design Language (AADL) [113]). To quantify the impact of unavailability of Supervisory Control And Data Acquisition (SCADA) systems, a mathematical method combined with mean failure cost (MFC) metric and the classic availability formulation is proposed [114]. A new set of solutions based on undirected graph is proposed to verify the topology verification and locate the failures in Software-Defined Networks (SDN) [115]. A dedicated symbolic model based on a combination of existing notions of robustness is introduced for designing robust CPS [116]. To overcome the incomparable of the dual conjunctive scheme, a method is developed for verifying N-inference diagnosability [117]. To evaluate the ability of withstanding and recovering from irresistible catastrophic, Alexander A. G introduced the concept of resilience and present two classes of formal models, which are multi-level directed acyclic graphs and interdependent coupled networks [118]. To formalize and assess the reconfigurable CPS, Linas L. et al. formally defined a dynamic system architecture in Event-B, and verified the correctness based on derived model. To guarantee the resilience of data processing, the authors transformed the Event-B model to the statistical UPPAAL and evaluated the likelihood of resilience under different system parameters with a statistical model checking [119]. To analyze the unpredictable and random behavior, the higher-order logic (HOL) is employed to model cyber-physical transportation systems formally. The authors obtained a randomized model by adding appropriate random variables and the probability or expectation of the system based on formal reasoning [120]. Taking TTA, FlexRay, and TTCAN as examples, Saha et al. analyzed the three startup algorithms of time-triggered architectures and presented specified formal models of faults for these architectures, then proposed different verification approaches for the three startup algorithms [65]. Compared with other two methods, formal behavior descriptive languages achieve a good balance between fidelity and performance. Consequently, we can build the model in almost the same detail with the general purpose programming language, and guarantee the correctness of the model by using formal evaluation.

### 4.2. Improving the Trustability of the Dependability V&V Results with Cross Validation

Currently, no language is powerful enough to model and analyze the behavior of SCPS at all levels. Moreover, with the increasing complexity of the models, it becomes more and more difficult to evaluate the correctness and the fidelity of models. To overcome these issues, we can apply *cross validation is necessary to check the correctness of simulation results*. Hence, integrated solutions become more and more popular. After reviewing 221 papers on model-based dependability V&V, we selected 14 papers (corresponding to 13 approaches) with relatively detailed introductions to their methods, which are shown in Table 2. There are three types of model integrated solutions, which are root-cause analysis (RAC), model to (formal) model (M2M) and model to simulation (M2S). For the RAC solution, several (traditional) formal models or dedicated models are applied independently to analyze the possibility of failures and the root causes behind the failure. Generally, the models of RAC work at different levels. M2M is an advanced RAC method. It just needs to build a meta-model, and then the toolset will automatically transform the meta-model to the RAC model. M2S solution also needs to build a meta-model then the meta-model will be generated into simulation models. Both M2M solution and M2S solution can improve the efficiency of modeling, as well as the consistency of models, which are meaningful for cross validation.

Dependability is an integrating property. As shown in Table 2, MDE-based solutions can be applied to evaluate the SCPS from various aspects, such as correctness, reliability and availability. To build a dependable SCPS, we should go a step further, comprehensively analyze the upper bound and the low bound of these indexes with considering the reorganization of architecture, and draw the scope of capability of different organizations, and build a straightforward mapping between situations and its most suitable organization. However, to our best of knowledge, there is no model or toolset, which is powerful enough to comprehensively model and analyze the whole SCPS. Currently, many conference papers are published to introduce the explorations on toolset development [134,135,136,137,138]. However, these works are at the beginning, it takes time to integrate tools and validate the correctness of model transformation. As how to develop a powerful MDE toolset is beyond the scope of this survey, we don’t discuss them in detail.

### 4.3. The Challenges of Model Based Dependability V&V

The general model transformation flow of the integrated solutions is shown in Figure 9. For an ideal integrated solution, designers just need to build the meta-model and set related platform- dependent properties, and then the toolchains can automatically finish the remaining work, such as generating the analytical models and simulation models, and generating the related dependability and weakness analysis repots. The MDE flow with an ideal toolchain is shown in Figure 10. To build a useful toolchain for SCPS dependability V&V, in addition to the tremendous development efforts, we still need to overcome four challenges. (1). *Specify the error behavior model*: developers should build a library of errors, sign all original sources of errors and define the trigger conditions and the occurrence probability of each faults; and for the safety-critical system evaluation, the consequence of failures should be given. (2). *Embed the specialized error behavior model into a computation-integrated model*: developers should bind the states and variables between normal and error models, define the connection of normal behavior and error behavior. (3). *Transform the dependable system model into RCA models or simulation models*, i.e., transforming the hybrid AADL behavior model to a Petri net model. (4). *Developing compositional libraries and supporting incremental validation.* SCPS contains massive (heterogeneous) subsystems. The costs to evaluate all possible combinations are too high. The compositional libraries are necessary and the models should support incremental validation. Only in this way can we efficiently verify SCPS with MDE solutions.

MDE is an ongoing research program to improve the quality of design and the efficiency of SCPS development. The main advantages of the MDE toolset is that it can automatically transform one meta-model to one or several detailed models, and then analyze the dependability with the current formal methods or simulation tools. Through formal model transformation, MDE can guarantee the consistency between meta-model and other analytical models, and improves the trustability of V&V results with cross verification. Models based on SCPS (dependability) V&V have become more and more popular [139,140,141]. MDE can help designers to find defects at design period and reduce the cost of development on the domain of the Internet of Things, People and Services (IoTPS) [45,142]. The exploratory studies have shown remarkable advantages in developing complex software for industrial applications [143]. However, MDE-based SCPS V&V is still at the beginning stage. Up to 30 August 2018, only four MDE toolchains have been proposed to improve the dependability of SCPS. One is about statistical model checking with considering stochastic delays and errors [144] and another three proposals focus on MDE-based debugging and testing [145,146,147]. As MDE is a vast independent field, the detailed discussion of MDE refers to the comprehensive survey [148]. The authors investigates 10 MDE approaches on high reliable/available component-based information and communication system from 13 characteristics. For another detailed survey on model-based root-cause analysis (RCA) readers may refer to [149], and for more basic introductions of MDE in embedded system development, refer to [150,151,152,153].

### 4.4. Brief Summary and Discussions

Models, especially formal dynamic models, play so important roles in CPS V&V and maintenance, that Lee even argued that “CPS theory is all about models” [154]. However, most of current V&V tools are closed source and have different input and output formats, and even incompatible parameter sets. A powerful MDE toolset to transform the meta-model into these V&V tools is lacking. Moreover, the model theory for dynamic system is just at the beginning, and powerful V&V tools are necessary to reduce the complexity of V&V. Generally speaking, probabilistic models are the mainstream for modelling the dependability of the dynamic system, i.e., Markov chain models [122], Bayesian networks [155], etc. However, it is infeasible to enumerate all available possibilities for a complex dependability model. The analysis results are doubtful if we don’t know the accurate probability of failures. Unfortunately, it is hard to obtain the accurate failure distribution of failures (*MQ1*).

With the increasing complexity of models and tools, how to guarantee the correctness of models and ensure the fidelity of simulation becomes a big challenge. Current modeling theories and simulation tools are unable to handle well the complexity of SCPS. To effectively evaluate the dependability of dynamic behavior (self-management), we should improve both modeling theory and simulation toolchains. Traditionally, we can describe the subsystems of SCPS with four types of models, which are discrete models (i.e., event-driven models), continuous models (i.e., control models), probabilistic models (i.e., the models analyzed with Monte Carlo simulations, such as Bayesian networks models) and deterministic models (i.e., physical models and deterministic automata). These models describe the system from different perspectives, but no single type of model can cover all perspectives. It seems impossible to comprehensively model the dynamic behavior of SCPS with single technology. While current integrated V&V tools can’t manage well the interactions between different models, there are two kinds of direction to overcome these issues: one is exploring the organic integration of multiple types of models (i.e., the exploration of co-simulation [156]). Another solution is building a meta-model, and transforming it into other proper analytical models (i.e., an AADL-based toolset [126]), and then improving the trustability of results by cross validation. To guarantee the correctness of models and reliability of simulations, we need to guarantee the consistency of model communication for the first approach, and guarantee the consistency of model transformation for the second solution.

Moreover, with the increasing amount of subsystems, the complexity of model analysis grows exponentially (i.e., state space explosion, which is a NP-hard problem). Improving C&C of subsystems’ model is the only available solution for this issue. An ideal MDE tool should not only be able to decompose the SCPS model into several submodels and analyze them separately, but also evaluate the fault propagation between submodels and compile the results without reanalysing the submodels. To support composition, the MDE tool should be able to evaluate the comprehensive effects of multi-error sources. With such a MDE tool, we can analyze the dependability and other qualities of the whole SCPS in a formal and compositional way.

## 5. The Safety of Self-Adaptation Strategies and the Dependability of Real-Time Decision Process

As SCPS can permanently change the physical world, we cannot emphasize too much the dependability of self-adaptation. Currently, only few studies have been published on the dependability V&V of the self-adaptation strategies for SCPS (i.e., only 11 results are returned with the keywords ((TS = “*Cyber Physical system*” or TS = “*industry 4.0*”) and (TS = “*self-adaptive*” or TS = “*self-adaptation*” or TS = “*self-adapting*”) and (TS = “*safety*” or TS = “*reliability*” or TS = “*dependability*”)) on Web of Science as of 30 August 2018. Among them, two papers are about “security” and one is patent). As the environment changes dynamical, the status information of cyber space and physical space has strong timeliness, SCPS should support *real-time self-adaptation* (self-management). However, few publications on real-time self-adaptive CPS are available. The investigation in this section will not be limited on CPS domain. The dependability of self-adaptation mainly involves two aspects, which are: (1) the safety and fitness of self-adaptation decision and (2) the dependability of decision process. In this section, we first briefly introduce the state of the art of self-adaptation methods. In Section 5.2, we discuss the solutions to make safe self-adaptation decisions. Then, we introduce the methods to guarantee the safety of decision processing at runtime in Section 5.3 and the dependability of real-time decision processing in the Section 5.4. Finally, a brief summary is made in Section 5.5.

### 5.1. Brief Overview of the State of the Art Self-Adaptation for CPS

Self-adaptation is a key technology to build the complex autonomic computing system, and we believe that self-adaptation will be the technological trend for CPS. However, the strategies for SCPS is just at the beginning, a few studies on SCPS had been published by 30 August 2018 (as seen in the last three rows in Table A1 in Appendix A). Theoretically, self-adaptation can be abstracted as dynamic optimization problems (DOPs). From the viewpoint of the main application levels in dynamic environments, the centralized self-adaptive algorithms can be classified into three types: applied at the metaheuristic level, applied at the mechanism for DOPs level, and applied at the combination level; and the conclusion of the survey shows that the self-adaptation algorithms with at combination level are significantly better than others [35]. However, the real-time constraint and decentralized self-adaptive algorithms haven’t been analyzed in this survey. More detailed introduction of the patterns of self-adaptation strategies can be found in [74,157,158], and surveys of swarm intelligence algorithms are presented in Refs. [159,160,161]. The statistics of references on self-adaptive CPS are investigated in Refs. [74,162]. As a detailed survey on the self-adaptation algorithms is beyond the scope of this paper, we just briefly introduce the self-adaptation strategies for SCPS in this Subsection.

As shown in Table 1, the current self-adaptation solutions include *rules-based methods* [46], and domain dedicated *knowledge-* [58] or *ontology* [163] -based methods. To develop these methods, one generally needs to spend a long time preparing the rule/knowledge base. Evidence-/case-based reasoning methods are generally interpreted as a supplementary solution. As SCPS has to face various situations, it generally needs a large rule/knowledge base, which is a big resource burden for SCPS, especially for battery supported subsystems. To overcome the problem, “AI” methods are proposed for different subsystems, such as *reinforcement learning*-based methods for actuators like robots [164,165,166], *swarm intelligence*-based methods for embedded subsystems [13,161] and *machine learning (ML)*-based methods for big data analysis [167].

“AI”-based methods, especially the statistical learning methods, need massive training data and huge computation resources. Currently, most of exploratory studies focus on big data driven decision making. E.g. Wang et al. predicted the influence changes of facilities over dynamic vehicles with Bayes method and trajectory-based Markov chain model [168]. Chen et al. introduced an objects searching system for IoT with context-aware hidden Markov model and ontology methods [163]. Martin M. et al. compared the effect of deep learning and four multi-class classifiers which include multiclass neural network, multiclass decision jungle, multiclass logistic regression and multiclass decision forest on data processing in Industry 4.0 [167]. Kush et al. redefined safety based on the terms of risk, epistemic uncertainty and harm for statistical machine learning [169]. Maier presented an appropriate passive online learning algorithm based on timed automaton for cyber-physical production systems [170]. To get good results and to avoid overfitting, a mass of high quality training evidence/data are needed [171]. Currently, these high quality evidences have been carefully pre-filtered and (manually) marked. The current applications are generally dedicated to server fixed scenarios. In the real world SCPS, elaborate data is generally not available. Meanwhile the amount of scenarios are very large and their bounds are ambiguous. In addition, the long training time and the high overhead limit the online applications of tiny sensors and actuators. The statistical results generally lack personalization, which may not fit well to the individual nodes in some cases, while CPS applications are location-aware and object-aware. However, no technology can simultaneously learn from both statistical data and historical data.

The research on “AI”-based self-adaptation decision making is far from mature, an enormous technical challenges still have to be overcome [19,162,172], especially the safety and correctness verification problem. As SCPS is safety-critical, it is important to guarantee the C&D of self-adaptation decision. However, most of “AI” methods are weak in interpretability. No solution can analyze the effect of each operation of “AI” decisions, which makes it difficult to verify the dependability of “AI” methods. What’s worse, it takes a long time to complete the whole self-adaptation loop (from sensing to making decisions and taking actions). The lagging decisions maybe no longer fit anymore the new situation. *Proactive and latency-aware self-adaptation* [173] is the only alternative solution. However, forecasting is a harder problem, no (“AI”) methods can effectively predict the future status of a complex system, such as the future environment, and the SCPS itself. Therefore, “AI”-based self-adaptation is not mature enough for practical SCPS, but it can be an assistive technology.

### 5.2. Improve the Fitness and Safety of the Prophetic Self-Adaptation Decisions and Strategies

Though many self-adaptation models and strategies have been proposed [17,19], the safety of self-adaptation is still an open issue. Considering the great value of SCPS, methods to guarantee the safety of self-adaptation decision are urgently needed, especially prophetic self-adaptation decisions [174,175,176]. Roughly speaking, we can estimate the fitness/safety of self-adaptation strategies from two perspectives during the decision-making period. One is calculating the safety with a fitness function and making safety aware self-adaptation decisions; another is verifying the safety requirements with V&V methods.

#### 5.2.1. Safety Aware Self-Adaptation Decision-Making

Generally, the safety can be quantified with the loss/risk if the decision doesn’t well fit the target context. Hence, we can apply risk assessment methods to estimate the safety of self-adaptation decisions. Massive numbers of methods and models are proposed in various domains to estimate the risk/loss if the decision fails [177]. However, risk assessment is still an open issue, because various factors affect the safety of a decision, such as the huge amount of influential factors, the incomplete evidence and data errors, etc. The traditional risk assessment solutions include rules-based methods, domain dedicated knowledge and ontology-based methods [178], which are highly dependent upon the a priori domain knowledge.

To overcome this issue, “AI”-based solutions can be applied. Similar to generic ML methods, we can define the risk functions as the (property) loss function. Mathematically, given the joint random variables X (features) and Y (labels) with probability density function $f_{X,Y}(x,y)$, and the mapping functions h∈H:X→Y (hypothesis), then we have the normal loss function L:H×Y→ℜ, the property loss function C:H×L→ℜ, and the expected value of loss/risk $R(H):\iint_{x,y}C((y,L(h(x),y))f_{X,Y}(x,y)dydx$, where L(h(x),y) is the discrepancy between the hypothesis h(x) and real value y and C(y,L) is the property loss if we accept the hypothesis h(y) and process the corresponding activities. The safety-aware ML methods search the best H with minimized R(H) and acceptable size |H|.

As a well-known conclusion, the accuracy of ML solutions depends on the quality of its training data. The main challenge of safety-aware ML methods is how to collect large amounts and high quality training data, especially in the cases that the failure data is scarce and the accurate property losses are difficult to evaluate. Moreover, the prophetic self-adaptation decisions are made based on the predicted statues. Besides the inaccurate prediction, it is also impossible to enumerate all statues and collect the corresponding a priori knowledge of loss (during training). Hence, in most of cases, H is incomplete and C is inaccurate. A priori knowledge integrated with ML methods may be a potential solution to reason in the cases data, and non-strictly quantitative loss function C:H×L→ℕ can be applied as a tradeoff solution to build a rough mapping.

#### 5.2.2. The Safety V&V Methods at Design Period and at Decision Making Period

The research on V&V of self-adaptive systems is a popular domain. Currently, many related surveys have been published. For example, a detailed statistical analysis on formal methods for self-adaptive systems is shown in [179]. The statistical results show that formal modeling languages with property specification is the first modelling choice and reasoning is the main research technology. Formal V&V methods are mainly used to verify efficiency, reliability, functionality and flexibility. Embedded system and service-based system are the top two application domains. Another survey comprehensively discuss the runtime verification for service-oriented systems in five categories: the logic and calculus oriented approaches, runtime workflow monitoring, state-based conformity assessment, aspect-oriented verification and SLA-driven compliance [180]. For each solution, the authors introduced the framework from eight selected characteristics, which include monitor, monitoring process, formal specification language, development language, monitoring data, realization mechanism, properties of interest and service composition. Then the authors conclude that developing dependable service-oriented systems need elaborately design of runtime verification process and self-healing and self-adaptation mechanisms.

Compared to the generic self-adaptive system, SCPS needs to support real-time self-adaptation and the time for V&V is limited. In some cases, these procedures should be executed in real-time. Besides the generic characteristics, the V&V of real-time self-adaptation should also verify the time related constraints, such as the BCET and WCTE, the convergence rate of strategies and recovery time of failures, as well as the time cost of verification. Currently, the V&V of real-time self-adaptation is a new topic. Only 18 papers (patents are excluded) are returned on Web of Science with the following keywords ((TS = “*real-time*” and (TS = “*evaluation*” or TS = “validation” or TS = “simulation”)) and (TS = “*self-adaptive*” or TS = “*self-adaptation*” or TS = “*self-adapting*”) and (TS = “*safety*” or TS = “*reliability*” or TS = “*dependability*”) not TS = “security”).

To automatically verify the safety of self-adaptation decision, SCPS should firstly specify the requirements of decision (processing), and then model the coming scenarios, then evaluate the fitness of decisions and hazards of decision processing under temporal constraints. To model the different decision requirements in different transportation scenarios, a self-adaptive requirements specification is proposed to instruct the behavior of intelligent transportation systems [181]. According to the given example, the specification includes both functional and nonfunctional requirements, and its format is similar to software development specification. The authors also mentioned that KAOS approach to model the adaptation goals. In the V&V domain, aiming at the temporal dependent correctness, a zone-based Time Basic Petri nets specification formalism is designed to model the timed self-adaptation behavior and to verify the timed robustness properties of self-healing [182]. The framework can divide the Petri nets into several zones to simplify verification of intra-zone properties, and apply inter-zone evaluation the global properties. Zone is a divide-and-conquer solution, which is a useful methodology to solve the state space explosion issue (*MQ2*). Another Petri net-based validation approach is proposed for speed synchronization of racing cars. This approach rewrites the safety properties and the Petri net model with a Z3 SMT solver, and automatically proves the safety properties in both design and implementation level [183]. To verify the time-bounded decision process, M.B et al. formalized the semantics of the adaptation logic of the component called OLIVE with constraint LTL over clocks and verified the properties with SMT-solver. The authors analyzed the upper bound of the temporal cost of both on-line and off-line verification. Further, the authors pointed out the need for ad hoc approaches to perform on-line verification and discouraged the use of general formalisms [184]. Generally speaking, these formal methods extend the generic solutions by attaching temporal constraints to self-adaptation models, and evaluate the safety of real-time adaptation by verifying the correctness and time bound of operations.

As the architecture plays an important role in self-adaptation, a comprehensive solution should not only verify the self-adaptation strategies, but also evaluate the fitness of applied architecture/topology and strategies, and the synergy between them. As a solution for *MQ3*, a systematic approach is proposed to deal with the self-adaptation under uncertainties and topology changes. The authors extended the MAPE-K loop with a rule-based structure knowledge base and a goal-based requirements knowledge base. On these basis, the authors also proposed a formal timed adaptation logic to build adaptation rules and a meta-adaptation layer to verify the adaptation rules with SystemC and UPPAAL timed automata. The meta-adaptation layer can evaluate the effect of previous adaptations and learn adaptation rules based on runtime models [185]. Ilias et al. proposed a goal-oriented design method for self-adaptive CPS, which is called Invariant Refinement Method for Self-Adaptation (IRM-SA). The IRM-SA traces the requirements between system and the distinct situations of environment, and supports the verification of the dependability of self-adaptation. To deal with operational uncertainty, the authors also introduced a predictive monitoring mechanism for ABSA system in their proof-of-concept demonstration [57].

Simulation based V&V methods are a useful solution to overcome the lack of possibility distributions (for *MQ1*). Compared to formal methods, it is easier to analyze the synergy between self-adaptation strategies and self-adaptive architecture. A stochastic game based method is presented to analyze the proactive latency-aware self-adaptation. The authors built a formal probabilistic model and verified the adaptation behavior with discrete event simulation [186]. Paolo et al. provided a formal rule based validation method for service-oriented applications and validated the behavior by scenario-based simulation [142]. Youngil et al. introduced a vehicle–driver closed-loop simulation method to verify the self-adaptation algorithm for integrated speed and steering control with the CarSim and MATLAB/Simulink [187]. A rule based self-adaptation algorithm are proposed for traffic light control, the authors analyzed the safety and liveness with Simulink [188]. Simulation-based methods can evaluate the safety by checking predefined rules/thresholds but also the statistics of unacceptable results. Obviously, simulation-based V&V methods can be regarded as a part of MDE solution, which suffer all the challenges that have been introduced in the Section 4.3. Considering the computation complexity of simulation, it is only possible to evaluate some subsystems in some specified scenarios.

Moreover, uncertainty is the main challenge of the V&V of the (prophetic) self-adaptation decision for future scenarios [189]. As a possible solution, we could integrate the risk assessment methods into V&V methods, and estimate the risk of mismatching between (prophetic) self-adaptation decisions and scenarios, especially the risk of mismatching caused by the environment changes during decision processing (for *RQ3* and *RQ4*). To deal with the uncertainties, one native idea is to determine the scope of self-adaptation strategies at the design period (with simulation methods), i.e., Building the set of all applicable situations where the fitness of self-adaptation is large than a given threshold or the R(H) is less than a threshold. Then SCPS estimates the variation range of future scenarios at runtime and verify the safety by checking the coverage of two scopes.

#### 5.2.3. Brief Summary and Discussion

The research on the safety V&V of real-time self-adaptation is just at the beginning. To develop safety V&V for SCPS, we should consider not only the issues in generic self-adaptive system, but also the temporal constraints, the adaptability of architecture/topology and the uncertainties, as well as the process time of V&V methods. Currently, formal solutions and simulations are the two main means applied in safety V&V. These two means can’t verify “AI”-based self-adaptation. To the best of our knowledge, there are only three conceptual discussions on the safety V&V for “AI” solutions. Sanjit et al. introduced a concept of trustworthy machine learning and argued to use end-to-end specifications and specifications mining to guarantee the safety of “AI” [174,175]. Zolt’an et al. introduced a reinforcement learning-based event-learning framework to improve the communication and control of SCPS, then discussed the problems of verification from context and goal [166]. From our point of view, we might be able to apply safetycases@run.time, simulation-based methods and even “AI”-based verification methods to verify the safety of “AI”-based self-adaptation at the stage of decision making. Meanwhile, we can apply runtime optimization methods and requirements@run.time methods to refine the decisions over time to accommodate changing environment. All in all, with the application of “AI”-based self-adaptation, safety V&V methods for “AI”-based self-adaptation need to be urgently studied.

### 5.3. Guarantee the Safety of Decision Processing with model@run.time Methods

As argued earlier, SCPS should make prophetic self-adaptation decisions to overcome the long delay. However, as the future is full of uncertainties, future prediction is still an unsolved problem, let alone reasoning the future status and making proper decisions. One direction to reduce uncertainty is by making short-term self-adaptation decisions. However, the shortest term depends on the minimum delay from sensing, making decisions to taking actions, which is limited by current physics technologies. Another is gradually refining the decision with model@run.time methods at runtime. Though we can’t predict the future, we can specify the applicable context for each self-adaptation decision and let the actuators decide whether to process the decision or not. Model@run.time is a systematic solution for prophetic self-adaptation, it can significantly reduce the unnecessary uncertainties [190]. Moreover, with model@run.time methods, SCPS can decompose the V&V procedure, all involved subsystems can verify the sub-decisions and refine them with newest observation under the guideline of requirement specification, which is a promising solution for real-time V&V.

Due to the potential capability to reduce the uncertainties, model@run.time methods are attracting more and more attention. The survey [191] introduced the model@run.time methods for open adaptive systems (OAS) and analyzed the state-of-the-art safety assurance solutions and challenges of model@run.time. The model@run.time methods were classified into four types: safetycertificates@run.time, safetycases@run.time, V&V model@run.time, and hazard analysis and risk assessment@run.time, where safetycase@run.time is a formalized, modular safety case, which can be interpreted and adapted at runtime. The safetycase@run.time methods attach the complete arguments to interpret the requirements of safe status and safe situations. A good safety case model has good C&C quality. The top-level safety goals can be decomposed into the detailed requirements for each subsystem. Therefore, the subsystems can dynamically check the safety goals with the safety requirements at runtime.

Mario et al. argued that “safetycases@run.time appears to be sufficient to support the assurance of a wide range of application scenarios of Open Adaptive Systems (OAS) in safety-critical applications” [191]. However, in our point of view, safetycases@run.time is not enough to guarantee the safety of SCPS. As analyzed in the Section 3.3.1, multi-term self-adaptation loops are necessary to overcome the communication delay and the shortage of resources. Requirements@run.time (with compositional decision) is necessary to instruct the cooperation among distributed subsystems, and requirements@run.time can also reduce the resources budget and uncertainties. Moreover, compared to OAS, SCPS has larger scale and more critical constraints on real-time use. Model@run.time of SCPS should not only consider the all challenges that OAS have to face (as seen in the Section 6.2 of the survey [191]), but also the real-time constraints, especially the performance of interpretation.

However, current model@run.time approaches are fragmented and dedicated to limited scenarios, no integrated safety assurance framework is available [191], which can’t well describe the multi-level architecture and model the dynamic behavior of SCPS. To model and verify continuous SCPS scenarios, model@run.time solutions should dynamically adjust the rules and cases, and to overcome the limited resources of embedded subsystem, we need to develop compositional models and rules, taking full advantage of the hybrid multi-role self-adaptive architecture, and build multi-level databases to cache the rules and cases. Moreover, the DSS should be able to recognize the scenarios and subsystems should be able to dynamically rebuild models to fit the scenarios.

As a kind of safety V&V method, the model@run.time solutions also suffer from the issues introduced in Section 5.2.2. Similarly, the research on real-time model@run.time is rare. No result was found in the web of science with (TS = *“model@run.time” and* TS = *“real-time” and* (TS = *“safety” or* TS = *“reliability” or* TS = *“dependability”))*, and only 26 papers are returned with the query “*(+real-time +self-adaptation+model@run.time) AND keywords.author.keyword: (+safety)*”. With further review, there are only two concept discussions [192,193] and two proposals [194,195] belonging to our topic. The two proposals take into account the deadline of adaptation actions [194] and the WCET of subsystems’ actions [195], and try to guarantee the safety of self-adaptation on timeliness.

### 5.4. Guaranteeing the Dependability of Real-Time Self-Adaptation (Decision Process)

SCPS is a large scale (geographically) distributed system. The self-adaptation loop takes time. Meanwhile, the cooperation between subsystems suffers from various uncertainties, i.e., *RQ2* and *RQ4*. To guarantee safety of self-adaptation, SCPS should decompose the decisions and send the sub-decisions to the proper subsystems. Thereafter the subsystems should cooperate with each other (with the coordination of local DSS) to guarantee the timeliness and dependability of sub-decisions process. In addition, SCPS should apply multi-level dependability aware scheduling, and be ready to take remedial measures when subsystems fail or the decision isn’t suitable any more. In this Subsection, we assume that SCPS have made correct and safe prophetic self-adaptation decisions, and focus on the dependability of self-adaptive decision processing.

#### 5.4.1. Guarantee the Dependability of Self-Adaptation with Multi-Object Optimization Methods

SCPS should process the decision at the right time and at the proper speed with as few resources as possible. Various factors can affect the progress of decision processing, such as, the unexpected failures and the uncontrollable jitters. To complete the decisions timely, efficiently and spontaneously, SCPS should take into account various runtime requirements, generate the optimal decision process plan, then select the proper executors, arrange redundant subsystems and set backup plans in case of failures. The problem of long-term plan making can simplify as the generic multi-object optimization problem (MOOP). Current optimization objects for SCPS include guaranteeing the real-time/deadline/timeliness [196,197] resource performance [198], dependability [73] and other functional objects [199]. Massive algorithms have been proposed for MOOP, such as evolution algorithms [200], a nature-inspired algorithms (i.e., ant colony and swarm optimization) [201]. As MOOP is a well surveyed topic, here, we will not analyze these algorithms in detail.

In case of short-term adaptation/scheduling, SCPS should schedule several self-adaptation procedures simultaneously, i.e., for self-driving cars, the system should adjust the speed to avoid collisions, plan its path to avoid congestion, and process the commands from passengers. The short-term self-adaptation/scheduling for SCPS is a dynamic multi-object optimization problem (DMOOP) [35]. Due to the real-time constraints, it generally cannot find the optimum solution for DMOOP in time. SCPS has to track the changing optimum with iteration. Moreover, the decisions process involves massive subsystems, whose failure rate is no longer ignorable, especially when the activities should be stated at the right time and finished in time. To archive dependability aware scheduling is necessary, SCPS should take into account the states of system and environment [202], fault tolerance scheduling [89] and energy-aware scheduling [203]. Moreover, most DOP algorithms are designed for centralized applications [35]. For distributed actuators, we need decentralized self-adaptation algorithms to coordinate interactions. A multi-threading multi-agent scheduling was proposed to search the efficient and fast solution of complex problems in real-time featuring rapid dynamic changes and uncertainty [204]. To process dependably, these algorithms should take advantage of design diversities, arrange several (heterogeneous) redundant subsystems in safety-critical scenarios.

#### 5.4.2. Guaranteeing the Dependability with Goal/Contract Based on Decomposable Self-Adaptation Decision (the requirements@run.time Approach)

SCPS involves multi-level scheduling [47,62], the scheduling strategies at each level have different optimization goals and different constraints, and some of them may even be conflicting. In current centralized architectures, subsystems generally know nothing about the global goals and constraints, they just try their best to respond to scheduling commands. As a result, subsystems cannot solve multi-object optimization problems under their own contexts, and could not verify the non-functional requirements of commands (or sub-decisions). To solve this issue, the requirements@run.time solution sends the sub-decisions with the corresponding requirements called contracts (as seen in Figure 7). With contracts, subsystems can know the global optimization goals and slightly adjust the plans to better fit the local contexts.

Following this idea, Jiang et al. first formally defined an event-based scheduling policy and proposed a decentralized scheduling method, this method can automatically decompose the scheduling policies into atomic scheduling policies and analyze those policies independently [205]. Ilias proposed a goal-driven decentralized self-adaptation solution as a proof of concept of IRM-SA [57]. IRM-SA allows capturing the compliance of design decisions with the overall system goals and requirements; and subsystem could verify the goals and requirements at runtime. Based on formally specifying requirements, Carlo proposed a runtime efficient parametric model checking solution to verify the dependability at runtime [129]. As a nonfunctional requirement, we also proposed runtime dependability decomposition and subsystem composition patterns with WCET constraints [73]. The key of goal/contract based on self-adaptation is improving the C&C of subsystems and decisions (as seen in the Section 3.3.2). Only in this way can subsystems be composed correctly and with dependability.

### 5.5. Brief Summary and Discussions of Dependable Self-Adaptation

The self-adaptation of SCPS needs the interplay between architecture and strategies. Hence, to guarantee the safety of self-adaptation, it needs to co-verify the behavior of architecture and strategies. The safety V&V methods should be applied at both the design period and runtime to overcome the resource limitations, the uncertainties and the real-time challenges. Currently, formal methods and simulation methods are applied to evaluate the correctness and safety of self-adaptation, and check the satisfiability, reachability and consistency of self-adaptation behavior in different scenarios. However, these methods generally is dedicated to specified scenarios, which can’t continuously verify the safety of SCPS. Moreover, it is still lack of safety V&V method for “AI” based self-adaptation. As SCPS contains multi-level self-adaptation loops, it is necessary to apply some methods, like model@run.time and goal/contract-based solutions to coordinate the self-adaptation at different level. We believe model@run.time is a useful solution for SCPS to reduce the uncertainty of V&V and achieve real-time safety evaluation.

## 6. Self-Healing Solution for SCPS

Dependable SCPS needs self-healing solutions to ensure its health, prevent failures and simplify the maintenance. However, self-healing methods will introduce extra complexity, which is harmful to achieve high dependability. To overcome the challenges of *RQ1* to *RQ6*, we should *use simplicity to control complexity* [29], elaborately employing both traditional methods and modern methods to reduce the side effects.

Several similar surveys on self-healing/maintenance have already been published. Santiago et al. first reviewed the existing maintenance principles for linear complex systems (LCS); through analyzing the similarities and differences between LCS and CPS, the authors gave the non-applicable, exportable and adaptation required principles for the CPS separately [206]. According to the concept of Maintenance 4.0 in Ref. [207], modern maintenance are mainly focused on self-healing, especially failure prediction.

In this section, we focus on the technologies of self-healing for SCPS. We first briefly summarized the traditional dependability methods applied in SCPS, and then focus on the modern methods of runtime self-healing. Following the classification of the means to achieve dependability, our investigation is organized into three parts: (1) fault tolerance, (2) fault prediction (forecasting) and prevention, (3) maintenance. The limitations of each method are also stated by considering the characteristics of SCPS.

### 6.1. Traditional Solutions to Improve the Dependability of Infrastructures

The theory and traditional means of dependability [102,103,104,105,106,107,108,109] have been well-studied for a long time. As the investigations listed in Table 1 indicate, the traditional dependability methods are still the mainstream choice to guarantee the dependability of SCPS, which get impressive results, especially in guaranteeing the dependability of infrastructures. As SCPS contains an abundance of subsystems (especially the cloud-based SCPS and MAS-based SCPS), *subsystem level redundancy* is a natural solution to improve system dependability. For cloud-based SCPS, VM-based resources isolation can significantly simplify the fault isolation approaches [45,47,49]. *Rule-* or *knowledge*-based fault diagnosis and reasoning show good results in fixed or rarely changed environments [59]. Expert systems are a systematic dependability solution [208]. Dependability *middleware* is also an alternative solution for heterogeneous subsystems, which can provide consistent behavior [36].

As SCPS is a holistic system, it needs to apply dependability means at all levels and all subsystems and protect the SCPS from different perspectives. Focusing on dependable hardware engineering, Sparsh et al. provided a comprehensive survey on reliability techniques for microarchitecture design (such as processor registers, cache and main memory [209]. From the time perspective, Gao et al. studied real-time signal-based *fault diagnosis* techniques and *fault-tolerate* techniques in the control domain [210]. From the data-centric perspective, Thaha et al. considered seven types of faults into account for, including offset faults, gain faults, stuck-at faults, out of bounds faults, spike faults, data loss faults and aggregation/fusion errors, and performed a qualitative comparison of the latest fault detection researches including centralized, distributed and hybrid algorithms in the WSN domain, and analyzed the shortcomings, advantages of those algorithms [211]. Focusing on the WSN domain, Samira et al. reviewed the *fault tolerance methods* at different WSN levels and classified these methods into four categories: power management, flow management, data management, and coverage/connectivity [212]. Numerous methods are available to guarantee the dependability of static architecture and predefined software operations. For systemic introductions to dependability/reliability engineering readers are recommended to consult several books [101,102,103]. It should be mentioned that all introduced dependability management belongs to the active fault handling measures. In some cases, “*let it crash*” maybe the best solution, because fault diagnosis may take too much time and dependability means may cost too much.

### 6.2. Modern Methods for Fault Tolerance

Fault tolerance is a key technology to guarantee the dependability of runtime decision processes. Through design diversity, SCPS contains numerous heterogeneous subsystems with isomorphic software (software has different implementation of the same logical function), which provides an ideal platform for applying the temporal-spatial redundancy mechanism at the subsystems level. Following the general process flow of fault tolerance, we organize the investigation into three parts: fault detection/diagnosis, fault isolation and fault recovery/tolerance.

#### 6.2.1. Data Driven Fault Detection/Diagnosis

Yang et al. focused on the fault of context inconsistency and surveyed the improvement techniques and the challenges of deploying dependable pervasive computing systems from the fields of context management, fault detection, and uncertainty handling [213]. The authors classified the detection methods into: (1) *statistical analysis-based detection*, (2) *pattern matching-based detection*, (3) *coverage criteria-based detection*, (4) *simulation based detection*. Yet most surveyed solutions just detect the inconsistency between the function context and the environment, and few of them detect/verify the inconsistency between the dependability management context and the status of system and environment. As mentioned in Section 2.4 (the schema (b) of self-healing), self-healing is also a common service of the system, which also needs self-healing. Unreliable detection may generate wrong diagnoses and make things worse.

From the academic viewpoint, the survey investigated the knowledge-based fault diagnosis and hybrid/active diagnosis [210]. Furthermore, the quantitative knowledge-based fault diagnosis can be divided into: (1) *statistical-analysis data-driven fault diagnosis*, (2) *nonstatistical-analysis data-driven fault diagnosis*, and (3) *joint data-driven fault diagnosis*. Similar to all “AI”-based method, the training data is the key to achieve the highest quality diagnosis result. However, (high quality) failure data is scarce because of the difficulty of collecting data and small failure rates. Moreover, data-driven diagnosis is good at recognizing the abnormal (unexpected) symptoms, but not at identifying real failure causes. As abnormal symptoms may be caused by normal behaviors, i.e., stochastic resource competition, or just in unrecorded small probability situations, without additional reasoning methods, the data-driven solutions will generate numerous false alarms and increase the false positive rate.

As knowledge-based fault diagnosis needs huge resources (memory and storage), it is generally applied in the cloud system (DSS). For sensors and actuators, lightweight detection/diagnosis methods are the mainstream solution, e.g., heartbeat methods, WCET-based detection, and threshold-/event-/state-based methods. Modern methods for embedded systems include: (1) *Model- based fault detection* [214,215], yet their flexibility is limited. (2) *Probabilistic fault detection and isolation* [216], which may increase the rate of false negatives. (3) *Cooperative fault detection* [217], through exchanging (checking) information between neighbors. (4) *Distributed fault detection* [218], based on the statistical result of the neighbors’ data. (5) *Machine learning-based fault detection* [219,220,221]; these methods are generally applied in the data center, and the performance depends on the quality of collected data and the complexity of tested system.

Similar to WSN, the faults of SCPS are local in the network. Hence, we can learn the fault detection strategies in WSN [211], and develop suitable strategies for SCPS. Compared to WSN, SCPS is location-aware and individual-sensitive, the data in each location is unique and may be irreplaceable for decision making, using the approximate value from neighbors will decrease the fitness of decision and increase the risk of failure (for *RQ2*). It needs to comprehensively apply statistics-based, time series-based and domain knowledge-based validation solutions to analyze the reliability of data.

#### 6.2.2. Virtualization Based Fault Isolation

Through isolating the resource and execution environment between applications, virtualization can isolate the faults at the subsystems level and naturally prevent the propagation of faults. For DSS, we can apply virtual machines (VM) or containers to isolate applications. For sensors and actuators, faults can be isolated by separate hardware, middleware [222] and container [223]. Virtualization can also be applied to improve the isolation of networks. A virtual platform called Xebra is proposed to provide isolated communication for SCPS [224]. Both node-level and network-level virtualization are studied for WSNs [225]. The authors classify the network virtualization into *virtual Network/Overlay-based Solutions, cluster-based solutions,* and *hybrid solution based on middleware or VM.* Besides the advantages summarized in Section 3.2, virtualization is a useful isolation solution and the snapshot of virtualization is also a convenient recovery technology. Meanwhile, migration technology enables to switch the workload from failed hardware and improve the availability of services.

#### 6.2.3. Modern Fault Recovery/Tolerance Solution

Though we can apply fault tolerance solutions at different levels, one things that should be kept in mind about SCPS is that suffers from the famous Byzantine failure. The Byzantine fault is any fault presenting different symptoms to different observers. As a networked parallel system built with unreliable (wireless) communication channels, Byzantine failure will destroy all fault tolerance solutions by undermining the consensus of observation. Thanks to the virtualization technology, an advanced 2f+1 replication solution is designed to achieve Byzantine fault tolerance [226]. Manuel et al. explored the role that diversity plays in defending faults and attacks through qualitative analysis, the result shows that diversity can tolerate the common-mode failures but will introduce undetected failures [68]. The effects (robustness) of redundant connections are studied through the simulation of six kinds of topology [227]. To tradeoff between the cost and reliability, a formal specification of spatial and temporal redundancy based reliability is proposed; and a framework is designed to determine the maximum reliability with a recursive formulation [228].

#### 6.2.4. Brief Summary of Fault Tolerance Methods

For SCPS, as all the redundant components suffer the same environment, redundancy with the same subsystems fails to deal with interference from the physical world. Diversity design with heterogeneous subsystems is the solution to tolerate these failures. To build a dependable SCPS without introducing too much complexity, we should take advantage of self-adaptive architecture by systematically adopting the spatial-temporal redundancy solutions and isolation at different levels. Following this idea, Cui et al. introduced a cross-layer fault tolerance method based on a self-adaptive architecture [229]. As self-healing solutions may also fail, especially for data driven detection/diagnosis, self-diagnosis may generate false alarms and trigger wrong recovery operations. Hence, it needs V&V methods to check the health of self-healing solutions, verify the alarms of self-diagnosis. Moreover, SCSP should also take into account the time cost of fault diagnosis and recovery, avoid resource competition with normal functions, and also SCSP should guarantee that the healing operations take effect before the system crash.

### 6.3. Modern Methods for Fault Prediction and Prevention

The ideal self-healing solution is predicting the faults and then preventing the failures. The fault prediction and prevention methods monitor the symptoms or abnormal behaviors, and evaluate the risk of faults, then prevent the failure with proactive measures, i.e., replacing the fragile component, or replacing it with safe actions, or rejecting (or freezing) a dangerous operation (i.e., rejecting accelerating a car at an unsafe distance). Fault prediction and prevention have been widely applied in the IoT domain. A fault prevention framework is proposed, which consists of monitoring, fault-prediction, and adaptation mechanisms [230]. To prevent selecting the unhealthy node as cluster heads, a fault-prevention clustering protocol is designed, which includes failure prediction, cost evaluation, and clustering optimization [231]. Deshan et al. proposed proactive systematic self-adaptation solution based on fault prediction and prevention to improve the reliability of mission-critical, embedded, and mobile software [232].

Technically, the key of fault prediction is monitoring the leading symptoms, such as the received signal strength indicator (RSSI) to battery power depletion [233] and the energy consumption state to estimate the remaining lifetime of a battery [234]. *A priori causal knowledge of failures is the selection guidance of leading symptoms, which can be gained from the traditional dependability solution*, i.e., expert systems [235] or reliability models [236] (i.e., AZOP and FMEA). Based on failure propagation models, Claudia et al. used the failure of error sources component as the leading symptom to predict the failure of error sink components, and guild the reconfiguration-based self-healing [237]. The detailed classification of online failure prediction methods is shown in Figure 11 [238].

As mentioned in Section 2.3, the self-management loop suffers non-negligible approximate errors and the probabilistic result for statistics-based fault prediction is another serious error source. The prediction results are generally expressed with a probability value whose domain is [0,1], while the execution of prevention and recovery operation is binary {0,1} (do or do not). For SCPS, mapping from [0,1] to {0,1} is incautious and risky. Proactive prevention may undertake unnecessary, even unsuitable, operations which increase the delay and waste resources, and even introduce new errors to the system. Through causal inference, *model-based* or *knowledge*-*based failure prediction & prevention* may address this issue. Furthermore, considering the complex error sources, current failure prediction methods are still not powerful enough to diagnose failures in real-time. *Long-term* and *short-term failure prediction* should be selectively implemented at different levels of SCPS according to the accessibility to the symptom set. Current prediction solution is monitoring one specified symptom for one (kind of) failure. Considering the complex of failures, more efficient symptoms should be researched urgently, thus we can use fewer symptoms to predict more failures, which can significantly simplify the failure prediction. What’s more, the accuracy of *prophetic failure prediction* should be improved, especially in unexpected situations. Moreover, as current fault prevention methods rarely consider their influence on normal behaviors and neighbor subsystems, we should verify the safety of these prevention operations and use multi-object dependability aware scheduling/arrangement (as seen in the Section 5.4.1) to coordinate failure prevention operations and self-adaptations, to achieve side-effect free self-healing (*RQ4*).

### 6.4. Simplify the Manual Maintenance

As an inference of the halting problem, self-healing solution can’t detect and heal all failures automatically and some accumulated failures can’t be recovered without external help, i.e., wearing out of hardware, and exhausted batteries. Therefore, manual maintenance is still needed. As SCPS contains massive numbers of subsystems, it is a big challenge to locate the failed nodes and replace them with an identical subsystem. Hence, we should redesign subsystems and make it be able to join the SCPS automatically, seamlessly without introducing additional interference (*RQ4*). Notice that the maintenance here represents manually recovering the subsystems from the failures that are unable to self-healing, i.e., replacing a broken engine for a self-driving car.

There are three main directions to simplify the maintenance. One is *simplifying the failure identification and location*. Maintainers can locate the failed nodes based on the reports from the neighbor nodes. Another technique is developing assistance programs to improve the efficiency of preventive maintenance. A *condition and time*-based maintenance model and framework is proposed and *domain knowledge*-based decision making for maintenance is highlighted [239]. The Bayesian networks-based assistance programs for dependability, risk analysis and maintenance is reviewed in [155]. A linear programming-based model is developed to optimize the repair priorities and a stochastic model is introduced to examine these reparation policies [240]. The third direction is simplifying repairing/replacing. The basic solution is designing the subsystem with high C&C, so the new subsystem can join into the SCPS seamless. As manual maintenance is the basic solution for traditional dependability engineering, here, we don’t discuss it in detail.

### 6.5. Brief Summary and Discussion

To improve the runtime dependability of CPS, comprehensive fault management methods should be elaborately integrated at different levels, and the side effects of self-healing methods should be evaluated at both design period and runtime. The means of developing a dependable SCPS are: (1) strengthening the reliability of infrastructures (hardware, software and network) by using the traditional methods (for *RQ1*); (2) improving the availability of sensor (network) and (network) with subsystem redundancy and reconstruction (for *RQ1*); (3) integrating fault detection/diagnosis methods (i.e., FDIR) on sensor nodes and actuator nodes to block error propagation and limit the failure effects (for *RQ1*); (4) enhancing the fault isolation at subsystem/application level (i.e., virtualization technology) (for *RQ1*); (5) using group intelligence (i.e., DFD, cooperative detection) to check with each other (for *RQ1* and *RQ2*); (6) applying data-driven methods (i.e., machine learning) on DSS to predict faults (for *RQ1* and *RQ5*); (7) building knowledge-based methods or expert systems on all subsystems to evaluate the correctness of prediction and then safety of prevention operations (similar to means for self-adaptation); the knowledge-based methods or exporter systems can take over control of self-healing system if the big data is unavailable (for *RQ1* and *RQ5*); (8) simplify manual maintenance and building human-in-loop SCPS (the backup solution for the halting problem).

Self-healing solutions can improve the dependability of the system, but they also increase the complexity and introduce new errors. Finding more efficient symptoms is one solution to reduce the complexity. To develop high performance, low side-effects self-healing solution, we should adopt these principles: (1) *keeping the fault management measures simple and verifiable*; (2). *taking the side effects of fault management into account* (e.g., time delay, resource budget, etc.); (3). If recovery costs are too high then subsystem can just *notify the stakeholders* (other subsystems and managers) and *let it crash safely*.

Current self-healing solutions are not powerful enough for SCPS. We can improve them in these fields: (1) *the timing order of events may be wrong* (*RQ2*), we can use a priori knowledge and cross validation to check the correctness of input data to improve the accuracy of data-driven solutions. (2) *Environment effects* are not considered in current studies. As the environment can directly affect the infrastructures of SCPS, especially the sensors and actuators, it will absolutely change the distribution and rate of faults. Fault prediction methods should take the environment effects into account. (3) *Human effects* are also not evaluated. The unpredictable and controllable human behaviors will be the new source of errors. It is urgently necessary to develop the theory to model the human behavior and predict the intention of human.

## 7. The Missing Pieces of the Technology Puzzle and the Future Directions of SCPS

SCPS is a rapidly progressing research area involving multidisciplinary technologies. It requires great skills to trade off among the complex requirements and integrate the fragmented technologies together to build an organic system. Moreover, to adapt to the dynamic uncertain environment, SCPS should balance the controllability and autonomy (as seen in the Section 3.1), automatically make proper self-management decisions with the incomplete, insufficient, even incorrect observations (*RQ2*), and coordinate subsystems to process the decision safely, dependably and timely in the changeable environment. To build dependable SCPS, we need systematic architecture designs, creatively integrate multidisciplinary technologies, and life-cycle V&V and continuous maintenance.

### 7.1. The Available and Missing Measures

To develop a dependable SCPS and to guarantee its dependability through the whole life cycle, we should comprehensively apply various technologies at different levels. Here, we summarize the popular (and potential) methods and frameworks from thousands of papers, classify them into four domains: correctness V&V methods, self-adaptation solutions and V&V methods, dependability and V&V methods, MDE solutions (toolsets), at five main levels: (1) hardware level (nodes and network devices), (2) basic software level (operating system and network protocols), (3) middleware and framework, (4) domain/aspect oriented solutions, (5) the all-in-one solutions. Considering the readability, we select part of the results, which are illustrated in Figure 12.

According to the jigsaw of technologies, we can draw eight conclusions: (1) the studies on precision timing behavior are just at the beginning stage, and most of current research focuses on the hardware and basic software level, and few systemic tools or theories are proposed. (2) The dependability methods can be applied at any level, while the weight of flexibility increases with the rising of level. (3) Simulation-based methods and dynamic logic-based models (whose analysis procedure is also simulation-based) dominate the V&V of correctness and dependability at the high level. (4) C&C is a promising solution to improve the flexibility and tame the increasing complexity. (5) Highly flexible self-adaptive architecture (i.e., ABSA, SDA, and SDN) is the cornerstone of SCPS. (6) It needs the co-design and cooperation of architecture and strategies to archive maximal and optimal adaptability. (7) MDE is a holistic solution to tame the complexity and the decrease the risk of CPS engineering. It should be mentioned that specification (8) is also a useful measure to guarantee the quality of CPS [241,242].

There are nine missing pieces in the jigsaw of current technologies: (1) theoretically, infrastructure and tools supporting for timing behavior at high level are urgently needed. (2) Precision-timed wireless networks are emerging. (3) Network integrated V&V and large scale supported toolsets are needed. (4) The coverage and performance of testing and simulation should be improved. (5) The side effects of dependability management should be continually investigated. (6) Dynamic multi-object optimization strategies are necessary. (7) Real-time V&V methods for the real-time self-adaptation are needed immediately. (8) Research on the cooperation of these dynamic strategies should be put on the agenda. (9) Safe “AI” methods and related safety V&V methods are necessary for data driven self-adaptation and fault predication [169].

### 7.2. Technical Challenges and Directions

SCPS contains various technologies. However, these technologies are fragmented and unstructured, which cannot meet the development and maintenance requirements of SCPS very well. In this section, we briefly summarize the challenges of dependable SCPS development, which are shown in Table 3. In addition, focusing on dependability, we analyze the urgency of related technologies from the perspectives of: (1) the maturity of the technology [6,39], (2) the social expectation and acceptance of technology [23], (3) the degree that other technologies depends on target technology (based on Figure 12). Notice that the challenges of security and traditional technologies are not discussed in this table.

### 7.3. Future Direction: a Concept of All-in-One Solution

SCPS should adjust its architecture and apply proper self-management strategies to achieve the best adaptability and efficiency. Considering the complexity and uncertainties, even we have improved the C&C of models, it is still impossible to completely evaluate the synergy between architecture and self-management strategies during design period. Model@run.time-based V&V will be the trend to guarantee the dependability of self-management.

However, current model@run.time approaches are fragmented and dedicated to specific scenarios, which cannot handle well the changeable requirements of SCPS and also they are unable to verify the system in unexpected situations. To evaluate the proactive self-adaptation decisions, the ideal model@run.time methods should be able to predict the future environment $S_{p}(t+\Delta t)$ without taking any self-adaptation actions; by evaluating the difference between the expected future $S_{p}^{e}(t+\Delta t)$ and $S_{p}(t+\Delta t)$ with the related domain knowledge, SCPS searches the proper self-adaptation actions $\Delta S_{p}^{n}(t)$ and self-healing actions $\Delta S_{c}^{m}(t)$ based on current status $S_{p}^{m}(t)$, and then generate the related requirement specification, such as timing orders and timeliness of self-management activities, the dependability and robustness requirements of services. To achieve this goal, we can integrate the contract-based solutions (as seen in the Section 3.3.2) and MDE/model@run.time-based solutions to build a multi-level simulation-based self-evaluation solutions, which is shown in Figure 13. The conceptual solution integrates a simulation-based decision making and evaluation, contract-based self-management, and multi-object optimization based (distributed) scheduling. This solution includes three main feedback loops. One loop is between global DSS and local DSS, the second loop is local DSS between sensors or actuators, and the third loop is between sensors, actuators and the physical world. Based on the three loops, SCPS can form a complete feedback loop between physical space and cyber space. The global DSS can simulate both the physical space and cyber space (especially the sensors and actuators), then select several best simulation results as the self-management decisions and generate the contract. The local DSS selects the advices/contracts best fitting to its situation, decomposes the contracts based on multi-object optimization, and coordinates subsystems based on multi-object scheduling with sub-contracts. The sensors and the actuators follow the sub-contracts and process self-management strategies in a decentralized way. This conceptual solution can also provide life-cycle dependability management, and improve the efficiency of development and traceability of management.

As current data analysis technologies can hardly learn the causality from data. Without causality, data-driven self-management is full of hazards. While formal models and a priori knowledge naturally contain the causality, which can guarantee the dependability of data driven self-management. With this conceptual solution, we can analyze the causality with the models and generate a holistic view of the causal relationship between events and effects beside on the behavior models, and the causal relationship between error sources and possible failures based on fault propagation models and data flows.

To realize this concept solution, we should develop MDE toolsets which can model both the cyber space and the physical world and the interplay between them. In addition, with the improvement of MDE toolsets, co-validation and co-simulation programs could be generated from the meta-models automatically. The simulator can evaluate the current situation and even forecast the future. For each change, the simulator can search the proper architecture and the optimal self-management strategies or a set of possible strategies (advices). The ideal MDE toolsets can also be applied to trace SCPS and collect data for self-learning, which forms a close-loop runtime self-evolvable V&V solution.

## 8. Discussions and Conclusions

Great wisdom is needed to develop and maintain a dependable SCPS. In this survey, we first briefly introduced the concepts of CPS, self-management and dependability, discussed the necessary of systematical design of dependable SCPS and elaborated integration of interdisciplinary technologies. We separately investigated and analyzed the current research on self-adaptive architecture design, self-adaptation strategies, self-healing solutions and the corresponding dependability V&V methods. The dependability of self-management in SCPS depends on the combined actions of the quality of self-adaptive architecture, the fitness and safety of self-management decision, the timeliness and reliability of decision processing, which needs both design period evaluations and runtime assessment. We believe that MDE and model@run.time will be the trends for V&V dependability at the design period and at runtime. Furthermore, we can integrate MDE and model@run.time to build all-in one solutions for future SCPS design and management.

The increasing complexity of SCPS and the uncertainty of the future environment are the two main error sources of dependability. To tame the complexity, the two principles are: (1) *using simplicity to control complexity* and (2) applying self-management to reduce the complexity of runtime management. Numerical technologies can reduce the complexity of designs, such as, formal models, standardization, abstraction and decoupling, etc. For reducing the complexity of self-management services, we can also apply software-defined architectures (SDAs) to archive self-adaptive architectures, decouple the control logic from normal functions, and improve the C&C of subsystems. As introducing new technologies will increase the global complexity, we need to systematically consider the side-effects of adopting new technologies, compromise between the complexity and flexibility/adaptability, and then elaborately and rigorously integrate these technologies.

To reduce the uncertainties, the self-management decisions should be gradual refined with the progress. In detail, the DSS can make prophetic self-management contracts, apply requirements@run.time methods to instruct the cooperation among the local subsystems and let the actuators make the final decision whether to activate or not. To deal with uncertainties, the local systems have enough autonomy to decide to accept or reject the contracts. They should also be able to refine process flow and related thresholds of the contracts based on their newest situations and observations.

Models play important roles in dependable SCPS design and safe self-management decision making. Considering the synergy, we should co-verify the dependability/safety of self-adaptive architecture and the self-management strategies, and check the best collaboration for different situations. As a systematical solution, MDE can improve the dependability from architecture and strategies design to the cooperation between teams and the results of validation. Model@run.time is another systematical solution to improve the dependability of self-management decision and the related procedure of decision process. With model@run.time solutions, SCPS can model the cyber space and physical space status, and then make the best suitable contracts and specify the requirements of contract execution, so that the distributed subsystems can process the contract dependably and timely. However, current MDE tools and model@run.time approaches are fragmented and dedicated to specified scenarios, so none of them can meet the requirements of systematical evaluation of SCPS.

Besides the traditional measures, we can also apply the newest technologies to improve the dependability, such as applying design diversity to improve fault tolerance, using virtualization technologies to simplify fault isolation and application scheduling, using middleware technologies to guarantee the reliability and consistency of operations among heterogeneous subsystems, dynamic multi-object scheduling to balance the different requirements (i.e., short-term requirements: timely, performance and dependability, and long-term requirements: energy and survivability). Moreover, we should take full advantages of the characteristics of SCPS, such as self-similarity of architecture and behavior and the massive (heterogeneous) redundant subsystems, to simplify the management and improve the dependability of SCPS., e.g. SCPS can use the transfer learning method to learn the model/behavior from similar subsystems, and the new joined node can learn from neighbors with the same role and initiate its own parameters. Meanwhile, the subsystem with same role can check the similarity of each other’s behavior to detect and prevent failures.

Domain knowledge, rules and expert system are the mainstream solutions to make the self-management decisions in current SCPS, and these solutions are also the main means to verify the safety and fitness of decisions at runtime. However, these methods are dedicated to predefined known scenarios, and can’t handle the unknown situations. To overcome this issue, more and more researchers are paying attention to “AI”-based or data-driven methods, and exploring the use these methods to build a self-learning system to improve the adaptability and intelligence of self-management decision making. However, the failure data (error behavior) is relatively scarce and difficult to collect, which also challenges the application of “AI”-based solutions to safety V&V. Moreover, most of “AI” methods are weak in interpretability, the related (safety) V&V methods are rare.

Moreover, as an inference of the Turing halting problem, it is impossible to build an absolutely dependable self-healing solution. Consequently, SCPS should estimate the cost of each self-healing decision, and let the subsystem crash safely in some cases. As the final backup healing solution, manual maintenance is still an essential option. Hence, we need to improve the observability and traceability of SCPS. Building such a user-friendly human-in-loop SCPS can also reduce the failure caused by human beings.

During the past decades, SCPS has become more and more popular. Researchers have explored numerous theories and tools to improve the quality of SCPS. However, these studies are fragmented. Considering the complexity of SCPS and MDE toolset, it needs the experts from different disciplines to build the all-in one solutions that will guarantee the dependability through the whole life-cycle of SCPS. Considering the huge workload of technology integration, interagency and cross-disciplinary cooperation are urgently necessary to promote research on dependable SCPS.

## Figures and Tables

**Figure 1 sensors-19-01033-f001:**
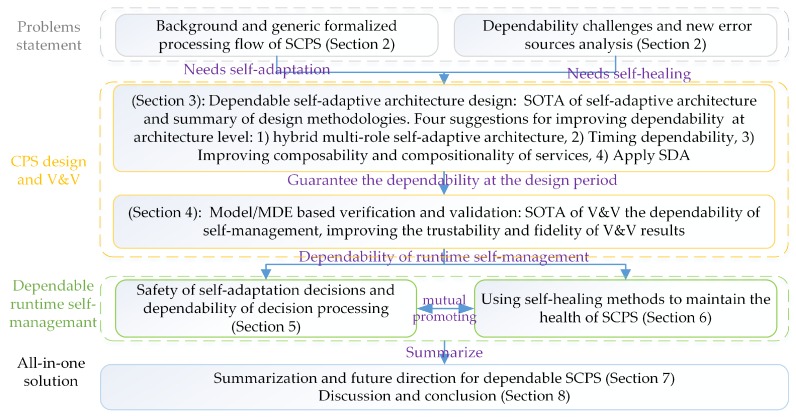
Structure and relationship of the survey.

**Figure 2 sensors-19-01033-f002:**
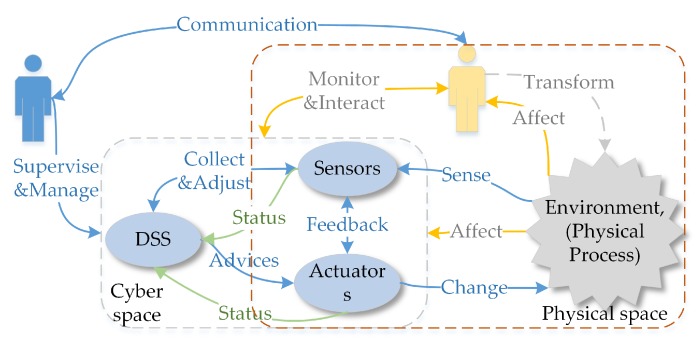
The generic process flow of environment-human-in-loop SCPS.

**Figure 3 sensors-19-01033-f003:**
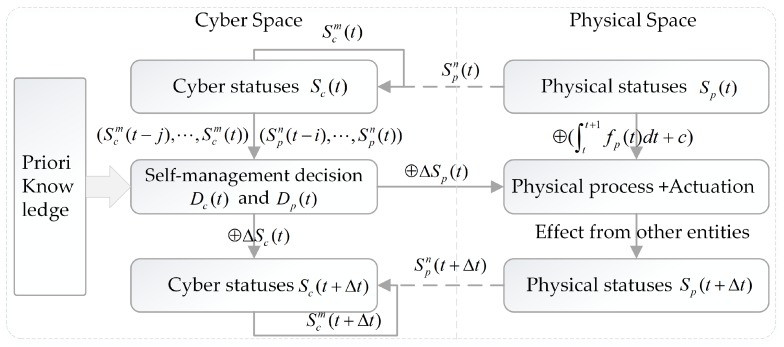
The formal interaction between cyber space and physical space.

**Figure 4 sensors-19-01033-f004:**
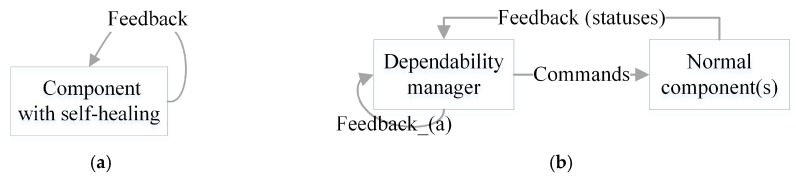
The schemas of self-healing. (**a**) Internal self-healing in a signal component; (**b**) External self-healing with the schema of diagnose-prediction-prevention.

**Figure 5 sensors-19-01033-f005:**
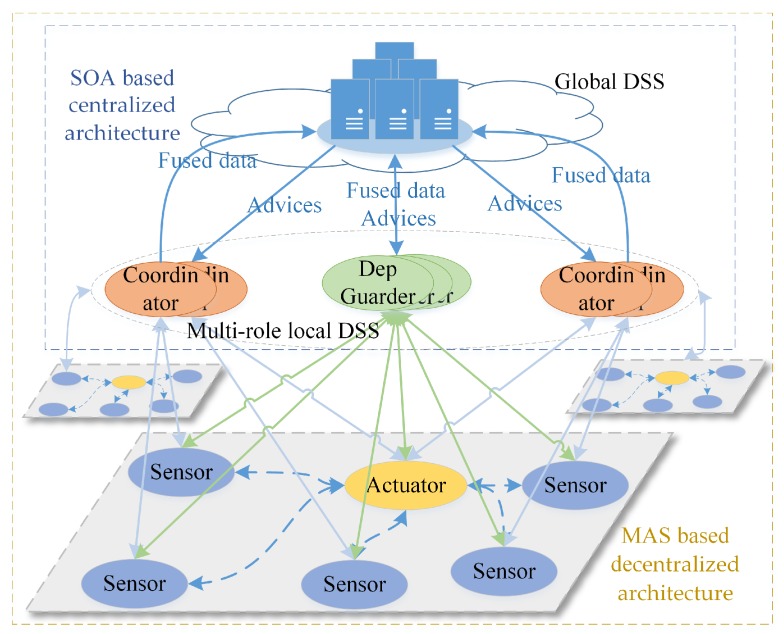
The concept of hybrid multi-role self-adaptive architecture.

**Figure 6 sensors-19-01033-f006:**
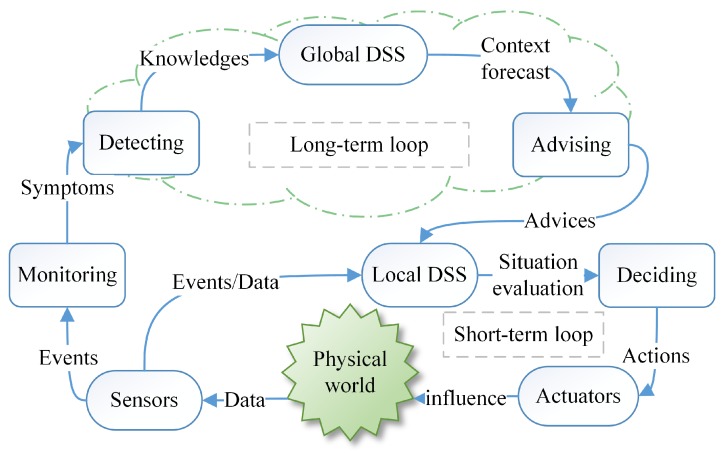
Multi-level and multi-term feedback loop [73].

**Figure 7 sensors-19-01033-f007:**
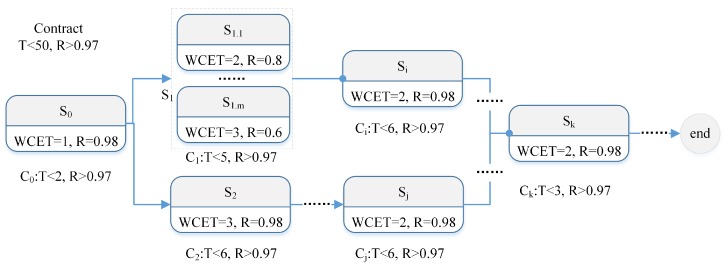
The concept model to guarantee the timing dependability with decomposable contract.

**Figure 8 sensors-19-01033-f008:**
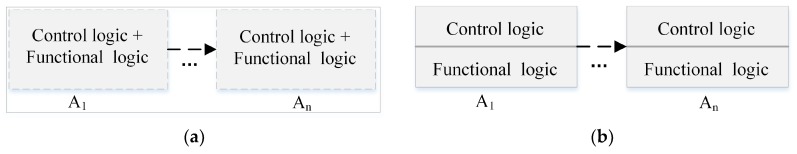
The schemas of decoupling: (**a**) monolithic design, (**b**) logic level decoupling, (**c**) role level decoupling with physical isolation. (*the arrow represents the direction of control logic flow).

**Figure 9 sensors-19-01033-f009:**
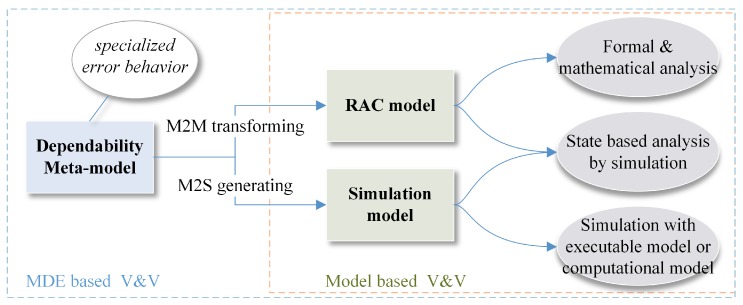
The general model transformation flow of model-based dependability V&V.

**Figure 10 sensors-19-01033-f010:**
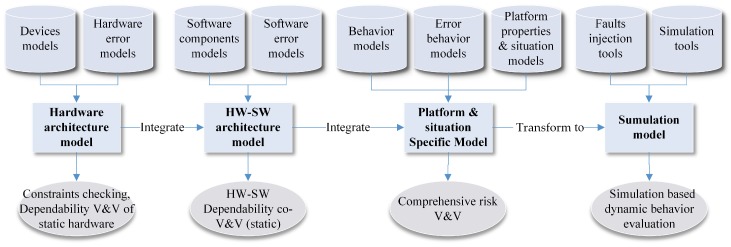
The concept solution of incremental dependability V&V for SCPS engineering: An AADL- based MDE view.

**Figure 11 sensors-19-01033-f011:**
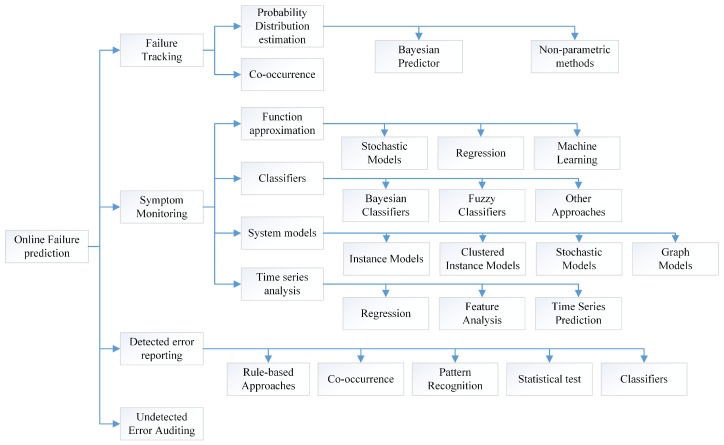
A taxonomy for online failure prediction approaches.

**Figure 12 sensors-19-01033-f012:**
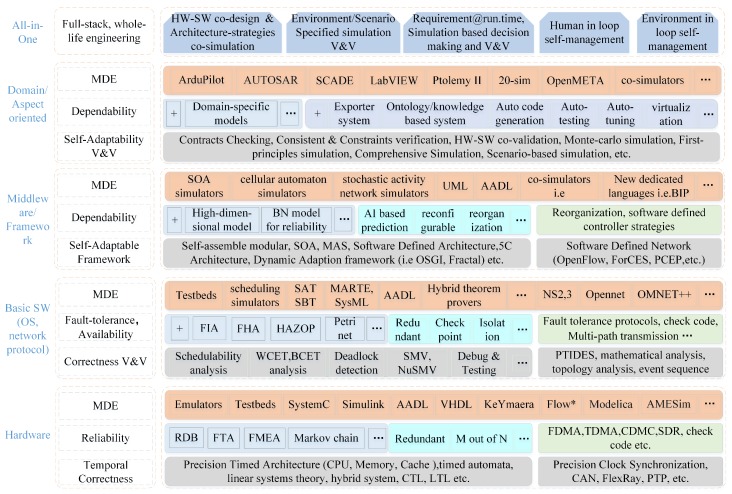
Available measures for dependable self-management CPS at different levels. (MDE is an integration of various technologies, the priority of technologies listed in the figure is theory > category of technologies > language > tool. The frequency is the main selection rule. Notice: As it’s impossible to list all related technologies in one figure, here, we just list some popular methods. Due to our limited knowledge, some other methods may be missed. All suggestions are welcomed to refine the jigsaw).

**Figure 13 sensors-19-01033-f013:**
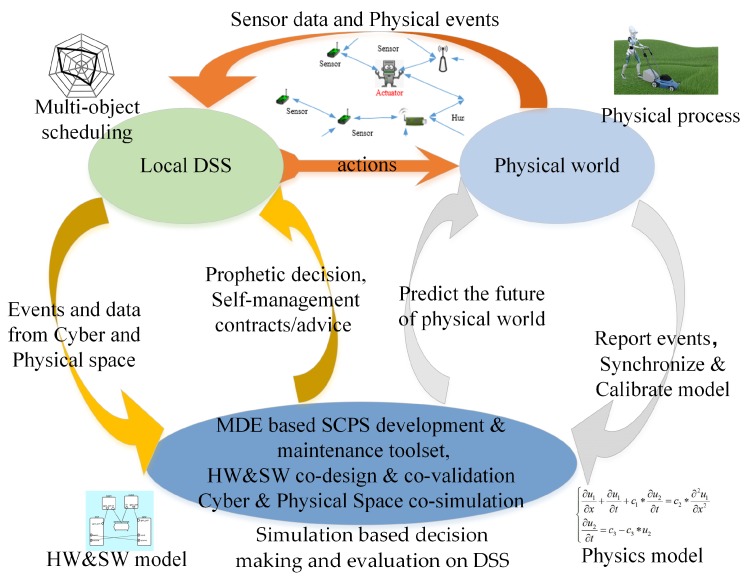
Co-validation and co-simulation through the whole life cycle of CPS.

**Table 1 sensors-19-01033-t001:** Current architecture studies on dependable SCPS.

Ref.	Arch	Low Complexity Self-Adaptive and Dependability ^1^ Means, etc
[36]	SOA based	Decouple, Compositional service; *Heartbeat, Real-time FDIR, Middleware based fault tolerate solution;*
[37]	SOA based	Unified Abstraction, Domain-Specific Description Schemas, Formal Semantics;
[38]	SOA based	Knowledge-Driven Service Orchestration, Ontology based service description;
[44]	SOA based	Formal contract for physical property, Dynamic physical behavior, Hybrid system behavior;
[45]	SOA based (ebbits)	Proxy, Virtualization, Middleware, Ontology, Semantic Knowledge, Rule base context recognition; *Predictive maintenance*
[14]	MAS based	Self-organizing & Self-adaptive models, Rules & Knowledge based Reasoning, proof-of-concept; *Exception Identification Model;*
[46]	MSA+Cloud	Data-driven self-organization, Intelligent negotiation based on contract net protocol, Deadlock prevention;
[41]	MSA+holons	Soft real-time MSA, Hard real-time function blocks (holons); *Redundancy;*
[47]	Cloud based	Virtualization, Multilevel smart scheduling algorithms; *Redundancy, checkpoints;*
[48]	Cloud based	Distribution middleware, Virtualized interrupt model, Spatial & temporal isolation based on partitioning; *Fault isolation;*
[49]	Cloud based	Virtualization, Task migration, Evolutionary algorithm for placement, WCET response time guarantee;
[50]	Software defined	Network-centered (SDN), Technology Standardization;
[40,51]	5C Arch	Decouple, knowledge based; *Prognostics and health management, Fault isolation* & *identification;*
[13,52,53]	Rainbow	Architecture-based self-adaptation (ABSA), (Re)scheduling, Strategy based, Mutation rules robustness tests;
[54,55,56,57]	DEECo	DSL, Decouple, IRM ^2^, Knowledge, Deterministic semantics, Formal analysis; *Proactive reasoning, Reliable communication*;
[58]	Na ^3^	Standardization, Open-Knowledge-Driven, Ontology;
[59]	Na	*8 steps comprehensive FDIR, Reliability Knowledge & Reasoning;*
[60]	Federation Arch	Component-based, Plug-in software, Plug-in runtime environment based on VM, Federation life-cycle management;
[61]	Na	*Fault mode, Reconfiguration, Rule based diagnosis, Reasoning;*
[62]	EVM ^4^	EVM, Virtual Component, EVM DSL ^5^, Formal design, Multi-level & multi-object scheduling

^1^ The methods for dependability are shown in *italics*; ^2^ IRM: Invariant Refinement Method; ^3^ Na: No introduction is available; ^4^ EVM: Embedded Virtual Machine. ^5^ DSL: Domain Specific Language.

**Table 2 sensors-19-01033-t002:** Recent studies on MDE/model-based SCPS dependability V&V.

Ref	Type ^1^	V&V ^2^	Key Analysis Technologies
[121]	RCA	R	Markov Chain (MC)
[122]	RCA	R	RBD, MC, Monte-Carlo simulation
[123]	RCA	R, A, ST	Stochastic Petri Net
[124]	RCA	D	MC, Stochastic Activity Network
[125]	M2M	D	Dependability domain ontology, FMEA
[126,127]	M2M	C, D	NuSMV, FTA, FMEA, HSIA, MC
[128]	M2M	C, A, Rb	BDD, BMC, FTA, FMEA
[129]	M2M	C, SF	Probabilistic temporal logic language, MC
[130]	M2M	R, A, M, SF	Bayesian Belief Network
[56]	M2S	SF, Rb	Simulation and statistical analysis
[131]	M2S	C, R	Automata-based diagnosis, LTL (Linear time Temporal Logic) based contract checking
[132]	M2S	C, Rb, R, SC	Calculation with mathematical model & Simulation
[133]	M2S	SF	Simulation and statistical analysis

^1^ RCA(root-cause analysis) means analyzing the dependability with original dependable model (without transforming); M2M (model to model transformation), which is the technology to transform the reference model to the RCA models; Sim is short for analyzing with original simulation model; M2S (model to simulation transformation), which means generating the simulation model from reference model; ^2^ Available items: Correctness (C), Dependability (D), Reliability(R), Availability(A), Safety (SF), Sustainability (ST), Robustness (Rb), Maintainability (M), Security (SC).

**Table 3 sensors-19-01033-t003:** The technical challenges of dependable SCPS.

Technical Area	Challenges and Directions	Urgency	Target
HW & SW infrastructure development	Precision timed, real-time HW & SW	High	Timing
	Standardization of subsystem (interfaces)	Medium	C&C
	Low power devices	Medium	Energy
Network communication & management	Precision timed network transmission	High	Timing
	Real-time (wired & wireless) communication	High	Timeliness
	Heterogeneous network management	Medium	Maintainability
Architecture design	Atomic service & subsystem design	Low	C&C
	C&C contract, interoperable subsystems	Medium	Self-*
	Discrete-continuous subsystem integration	Medium	Correctness
	Invariant behavior of integration	High	Correctness
	Theory for dynamic architecture	High	Flexibility
	Design methodology for dependable SCPS	Medium	Complexity
Middleware	FDIR middleware & Node level self-healing	Medium	Dependability
	Light-weight virtualization & migration	Medium	Self-*
	Domain ontology, Knowledge database	Medium	Self-*
	Service discovery & combination	High	Self-*
Consistent spatial-temporal & context cognition	Global reference time for large scale CPS	High	Timing
	Low cost clock synchronization	Medium	Correctness
	Global location reference for mobile CPS	Low	Correctness
	Consistent data and context assurance	High	Correctness
Lifecycle management (self-management)	Manage dynamic & changeable architecture	High	C&D
	Multi-objective (prophetic) adaptation	High	C&D
	Knowledge-driven decision making	High	C&D
	Decision/adaptation safety/evaluation	Medium	Safety
	Situation aware self-healing & notification	High	Dependability
	Causality analysis	High	C&D
	HMI for human-in-loop CPS	High	Usability, safety
Modeling & validation & MDE tools	Dynamic architecture modeling	High	Fidelity
	Multidisciplinary modeling	High	Modeling
	Consistent of model transforming	High	Correctness
	Evaluation the correctness of models	High	Correctness
	Holistic modeling theory or methodology	Medium	Modeling
	Situation based model V&V	Medium	V&V
	MDE toolchains (design, V&V, coding, testing) and life cycle V&V supporting	Medium	Consistency & efficiency
Simulation	Discrete-continuous-probability model co-sim	Medium	V&V
	Holistic multidisciplinary simulation	High	V&V
	Environment-in-loop simulation	Medium	V&V
	Human-in-loop simulation	High	V&V
	Fidelity evaluation	High	Correctness

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
