# Peer review of "A Comprehensive Technological Survey on the Dependable Self-Management CPS: From Self-Adaptive Architecture to Self-Management Strategies"

_sensors, 2019, doi:10.3390/s19051033_

Round 1

Reviewer 1 Report

This paper describes the results of analysis and survey on various technologies such as concept, architecture, V & V for dependable self-management CPS. The contents are extensive and systematic, but the following shortages must be supplemented.

1) It is difficult to find out what the focus to be claimed is on this paper. Survey papers, generally, are limited to a specific subject area, and provide meaningful information through in-depth analysis. However, this paper is expansive because it includes six topics, but the analysis depth of the survey is rather lacking. The purpose of this survey is not clear.

2) This manuscript looks like a research report, rather than research paper.

3) The expression of terms in the contents is inconsistent, for example, runtime /run-time.

4) In Line # 143, the problem of bottleneck was mentioned, but need a specific explanation of what bottleneck is.

5) There is an invisible characters in Figure 1. Modification is required.

6) Figure 2 differs from the explanation within the main text. In the text, Figure 2 was described as SCPS, but the title of Figure 2 is CPS. Also, it is not clear where the self-management part is included in Figure 2.

7) In the contents of Figure 3,

  (7.1) Line #281:   If 'A' is activity, then {A} must be a set of activities. However, {A} is defined as cyber space statuses in the paper, so it is difficult to understand semantically.

  (7.2) In the main text, D_p (t, p) is defined as a self-adaptation decision, and rectangle box named with ‘Self-adaptation decision’ is also defined in Figure 3. Not semantically understandable. The rectangle is ambiguous as to whether it is an activity or an output. The contents of the research paper should be clear. It seemed as a violation of the formal expression of research paper due to give a duplicate name to two different things.

8) in Line #299,  High-level rules, Priori Knowledge : There is no explanation for these terms, and it is difficult to understand. It is necessary to explain what these mean and what these are examples of.

9) Numbering errors: Section numbers in 5.4, 5.3.1, and 5.3.2 should be modified.

10) As expressed in the title of this paper, this paper is on the SCPS surveys and their analysis. However, by describing all issues related to the CPS/providing a comprehensive overview of all relevant topics, this paper includes all indirectly related classical topics such as distributed processing systems, real-time systems, and safety analysis techniques.

CPS corresponds to the technology areas that have to include both cyber space and physical space. Therefore, it may be necessary to mention about the technology that is directly related to the CPS.

Author Response

We would like to thank the reviewers for careful and thorough reading of this manuscript and for the thoughtful comments and constructive suggestions, which help to improve the quality of this manuscript. Please find the detail answer in attached file. Thank you for your time very much.

Reviewer 2 Report

p.p1 {margin: 0.0px 0.0px 0.0px 0.0px; font: 12.0px 'Helvetica Neue'} p.p2 {margin: 0.0px 0.0px 0.0px 0.0px; font: 12.0px 'Helvetica Neue'; min-height: 14.0px} li.li1 {margin: 0.0px 0.0px 0.0px 0.0px; font: 12.0px 'Helvetica Neue'} ul.ul1 {list-style-type: hyphen}

The survey concepts presented in this paper targets the concepts of CPS, self-management, and dependability, discussed the necessity of systematical design of dependable SCPS and the lack of comprehensive survey on the technologies integration of SCPS. Then, they separately investigated and analyzed the current researches on self-adaptive architecture design, self-adaptation strategies, self-healing solutions, and the corresponding dependability V&V methods. The paper is very well organized. Some minor suggestions are in order:

Can you describe the existing method addressed informal availability and reliability of dependable SCPS? 

Can you highlight some recent AI-based CPS addressed the SCPS?

Author Response

We would like to thank the reviewers for careful and thorough reading of this manuscript and for the thoughtful comments and constructive suggestions, which help to improve the quality of this manuscript. Thank you for your time very much.

Reviewer 3 Report

The article presents CPS concepts, self-governance and reliability, the need for systematic design of a reliable SCPS. The study discusses the technical trends of reliable CPS design and maintenance. A comprehensive solution was proposed to integrate technologies and create a reliable organic system. Current research on self-adaptive architecture design, self-adaptation strategies and appropriate V & V reliability methods has been described. The problem and research are very widely described on the basis of the collected data. Lack of sufficient hard information (scientific research) about the quality of credibility and quality of these data.

Author Response

(The authors gave the same response as above.)

Round 2

Reviewer 1 Report

Even though the contents of the paper is vast, they ware systematically composed and analyzed the existing studies. Although the contents were complemented through the revision and the intention of the survey was revealed, it is regrettable that this paper cannot be focused on just only SCPS area.

Nevertheless, this paper will definitely help those who are studying the CPS field, thanks for authors’ efforts in writing and analyzing this survey paper.

Please, check that the list of authors’ name and their affiliations are correctly mapped.